# DS-STAR: DATA SCIENCE AGENT VIA ITERATIVE PLANNING AND VERIFICATION

## ABSTRACT

Data science, which transforms raw data into actionable insights, is critical for data-driven decision-making. However, these tasks are often complex, involving steps for exploring multiple data sources and synthesizing findings to deliver insightful answers. While large language models (LLMs) show significant promise in automating this process, they often struggle with heterogeneous data formats and generate sub-optimal analysis plans, as verifying plan sufficiency is inherently difficult without ground-truth labels for such open-ended tasks. To overcome these limitations, we introduce DS-STAR, a novel data science agent. Specifically, DS-STAR makes three key contributions: (1) a data file analysis module that automatically explores and extracts context from diverse data formats, including unstructured types; (2) a verification step where an LLM-based judge evaluates the sufficiency of the analysis plan at each stage; and (3) a sequential planning mechanism that starts with a simple, executable plan and iteratively refines it based on the DS-STAR's feedback until its sufficiency is verified. This iterative refinement allows DS-STAR to reliably navigate complex analyses involving diverse data sources. Our experiments show that DS-STAR achieves state-of-the-art performance across three challenging benchmarks: DABStep, KramaBench, and DA-Code. Moreover, DS-STAR particularly outperforms baselines on hard tasks that require processing multiple data files with heterogeneous formats.

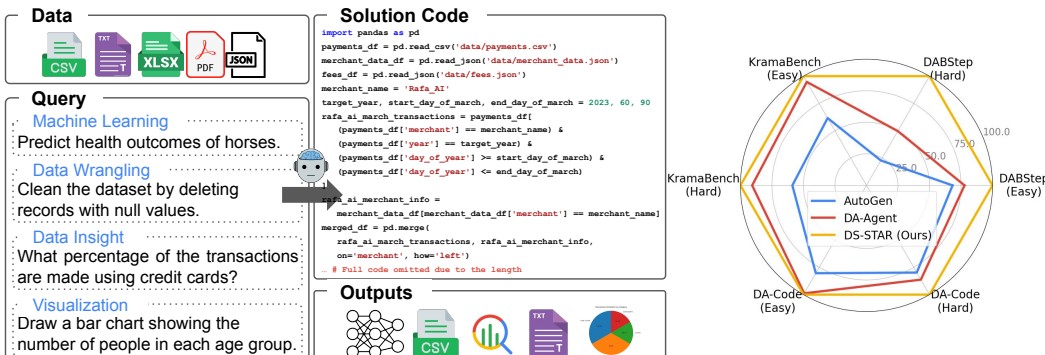

**(a) Data Science Tasks**          **(b) Experiment Summary**

Figure 1: (a) Data science tasks require processing data files in various formats (*e.g.*, csv, txt, xlsx, md) to answer user queries, such as data analysis for extracting useful insights or predictive tasks using machine learning. Data science agents are designed to accomplish this by writing code scripts (*e.g.*, Python) and answering based on the output from their execution. In addition to the solution code, the output may include a trained model for prediction tasks, processed databases, text-formatted answers, visual charts, and more. (b) We report normalized accuracy (%) for both easy (answer can be found within a *single* file) and hard (requires processing *multiple* files) tasks in the DABStep, KramaBench, and DA-Code benchmarks. DS-STAR consistently and significantly outperforms competitive baselines, particularly in challenging hard tasks, showing DS-STAR's superiority in handling multiple data sources in heterogeneous formats.

# 1 INTRODUCTION

The data science, which transforms raw data into actionable insights, is critical to solving real-world problems (De Bie et al., 2022; Hassan et al., 2023); for instance, businesses rely on insights extracted from data to make crucial directional decisions (Sun et al., 2018; Sarker, 2021). However, the data science workflow is often complex, requiring deep expertise across diverse fields such as computer science and statistics (Zhang et al., 2023; Hong et al., 2024). This workflow involves a series of time-consuming tasks (Zhang et al., 2024; Martin Iglesias, 2025), ranging from understanding distributed documents to intricate data processing and statistical analyses (see Figure 1).

To simplify this demanding workflow, recent research has explored using large language models (LLMs) as autonomous data science agents (Hong et al., 2024). These agents aim to translate natural language questions directly into functional code that generates corresponding answers (Ouyang et al., 2025; Wu et al., 2023; Yang et al., 2025; Yao et al., 2023). While representing significant advancements, current data science agents face several challenges that limit their practical utility and impact. A primary limitation is their predominant focus on well-structured data, such as relational databases composed of CSV files (Pourreza et al., 2024; Yu et al., 2018; Li et al., 2024). This narrow scope neglects the wealth information available in the heterogeneous data formats encountered in real-world scenarios, which can range from JSON to unstructured text and markdown files (Lai et al., 2025). Furthermore, as many data science tasks are framed as open-ended questions where ground-truth labels do not exist, verifying an agent's reasoning path is a non-trivial challenge. For example, most agents, like Data Interpreter (Hong et al., 2024), terminate their process upon successful code execution; however, executable code does not always guarantee a correct answer. Consequently, this lack of robust verification often leads an agent to adopt a sub-optimal plan for solving the given task.

To address these limitations, we introduce **DS-STAR**, a novel agent that tackles data science problems by generating solution code through a robust, iterative process of planning and verification. The proposed DS-STAR framework consists of two key stages. First, to ensure adaptability across diverse data types, DS-STAR automatically analyzes all files within a given directory, generating a textual summary of their structure and content. This summary then serves as a crucial context for the DS-STAR's approach to the given task. Second, DS-STAR enters a core loop of planning, implementation, and verification. It begins by formulating a high-level plan, implementing it as a code script, and then verifying to solve the problem. Notably, for verification, we propose using an LLM-based judge (Gu et al., 2024; Zheng et al., 2023) that is prompted to explicitly evaluate whether the current plan is sufficient to solve the problem. If the verification fails (*i.e.*, the judge determines the plan insufficient), DS-STAR refines its plan by adding or modifying steps and repeats the process. Crucially, rather than creating a complete plan at once, our agent operates sequentially, reviewing the results from each intermediate step before building upon them. [1] This iterative loop continues until a satisfactory plan is verified, at which point the corresponding code is returned as the final solution.

We validate the effectiveness of our proposed DS-STAR through comprehensive evaluations on a suite of established data science benchmarks. These benchmarks, including DABStep benchmark (Martin Iglesias, 2025), KramaBench (Lai et al., 2025), and DA-Code (Huang et al., 2024) are specifically designed to assess performance on complex data analysis tasks (*e.g.*, data wrangling, machine learning, visualization, exploratory data analysis, data manipulation, etc) involving multiple data sources and formats (see Section 4). The experimental results demonstrate that DS-STAR significantly outperforms existing state-of-the-art methods across all evaluated scenarios. Specifically, compared to the best alternative, DS-STAR improves accuracy from 41.0% to 45.2% on the DABStep benchmark, from 39.8% to 44.7% on KramaBench, and from 37.0% to 38.5% on DA-Code.

The contributions of this paper can be summarized as follows:

- We introduce DS-STAR, a novel data science agent which operates on data with various formats and on various tasks like data analysis, machine learning, visualization, and data wrangling, etc.

- We propose an LLM-based verification module that judges the sufficiency of a solution plan.

- We develop an iterative refinement strategy, enabling DS-STAR to sequentially build its solution.

- We provide a comprehensive evaluation on challenging benchmarks, showing DS-STAR's efficacy.

---

[1]This mirrors the interactive process of an expert using a Jupyter notebook to observe interim results.

## 2 RELATED WORK

**LLM agents.** Recent advancements in LLMs have spurred significant research into autonomous agents designed to tackle complex and long-horizon tasks. A key strategy employed by these multi-agent systems is the autonomous decomposition of a main objective into a series of smaller, manageable sub-tasks, which can be delegated to sub-agents. General-purpose agents, such as ReAct (Yao et al., 2023), HuggingGPT (Shen et al., 2023), and OpenHands (Wang et al., 2024), utilize external tools to reason, plan, and act across a wide range of problems. Building on these foundational capabilities, research has increasingly focused on specialized domains such as Voyager (Wang et al., 2023) for navigating the Minecraft environment and AlphaCode (Li et al., 2022) for performing sophisticated code generation. Furthermore, DS-Agent (Guo et al., 2024), AIDE (Jiang et al., 2025), and MLE-STAR (Nam et al., 2025) are tailored specifically for machine learning engineering. In a similar vein, our work, DS-STAR, is a specialized LLM agent for data science tasks.

**Data science agents.** Recent research has focused on developing data science agents that leverage the advanced coding and reasoning capabilities of LLMs (Jiang et al., 2025; Jimenez et al., 2024). Initial efforts utilized general-purpose frameworks like ReAct (Yao et al., 2023) and AutoGen (Wu et al., 2023). Following these pioneering approaches, agents like DA-Agent (Huang et al., 2024), which are specialized in autonomously generating code-based solutions for tasks such as data analysis and visualization (Hu et al., 2024; Jing et al., 2025), has been developed. A notable example is Data Interpreter (Hong et al., 2024), which employs a graph-based method to decompose a primary task into a series of manageable sub-tasks and the resulting task graph is then progressively refined based on the successful execution of these sub-tasks. However, a key limitation is that relying solely on successful execution as feedback often results in sub-optimal plans, since such feedback cannot confirm the plan's correctness. To address this, DS-STAR introduces a novel verification process that employs an LLM as a judge (Gu et al., 2024) to assess the quality of the generated solutions.

**Text-to-SQL.** The task of Text-to-SQL involves translating natural language questions into executable SQL queries (Li & Jagadish, 2014; Li et al., 2023). Early approaches relied on sequence-to-sequence architectures, which jointly encoded the user's query and the database schema (Cao et al., 2021; Choi et al., 2021). However, after the rise of LLMs, initial research focused on prompt engineering (Pourreza & Rafiei, 2023). This was soon followed by the development of more sophisticated, multi-step pipelines incorporating techniques such as schema linking, self-correction, and self-consistency to enhance generation accuracy (Talaei et al., 2024; Pourreza et al., 2024; Maamari et al., 2024). While these pipelines function as specialized agents for data analysis, their fundamental reliance on generating SQL restricts their use to structured, relational databases. In contrast, DS-STAR addresses this limitation by adopting a Python language, thereby framing our approach as a Text-to-Python task. To further facilitate this, we introduce a novel data file summarization mechanism which enables our method to demonstrate significantly broader applicability than traditional Text-to-SQL systems, as it can operate on a wide array of data formats, including JSON, Markdown, and unstructured text.

## 3 DS-STAR

In this section, we introduce DS-STAR, a framework for data science agents that leverages the coding and reasoning capabilities of LLMs to tackle data science tasks. In a nutshell, our approach first analyzes heterogeneous input data files to extract essential information (Section 3.1). Subsequently, DS-STAR generates a code solution through an iterative process (Section 3.2). The prompts and algorithms are detailed in Appendix H and A, respectively.

**Problem setup.** Our goal is to automatically generate a code solution $s$ (*i.e.*, a Python script) that answers query $q$ using a given data files $\mathcal{D}$. We formulate this as a search problem over the space of all possible scripts, denoted by $\mathcal{S}$. The quality of any script $s \in \mathcal{S}$ is evaluated by a scoring function $h(s)$, which measures the correctness of its output, $s(\mathcal{D})$, against a ground-truth answer (*e.g.*, accuracy) where $s(\mathcal{D})$ represents the output generated by executing $s$, which loads the data $\mathcal{D}$ to produce an answer. The objective is to identify the $s^*$ that is: $s^* = \arg\max_{s \in \mathcal{S}} h(s(\mathcal{D}))$.

In this paper, we propose a multi-agent framework, $\mathcal{A}$, designed to process the query $q$ and data files $\mathcal{D}$, which may be numerous and in heterogeneous formats. This framework is composed of $n$ specialized LLM agents, $\{\mathcal{A}_i\}_{i=1}^n$, where each agent possesses a distinct functionality, as detailed in the subsequent sections (see Figure 2 for the problem setup and our agent's overview).

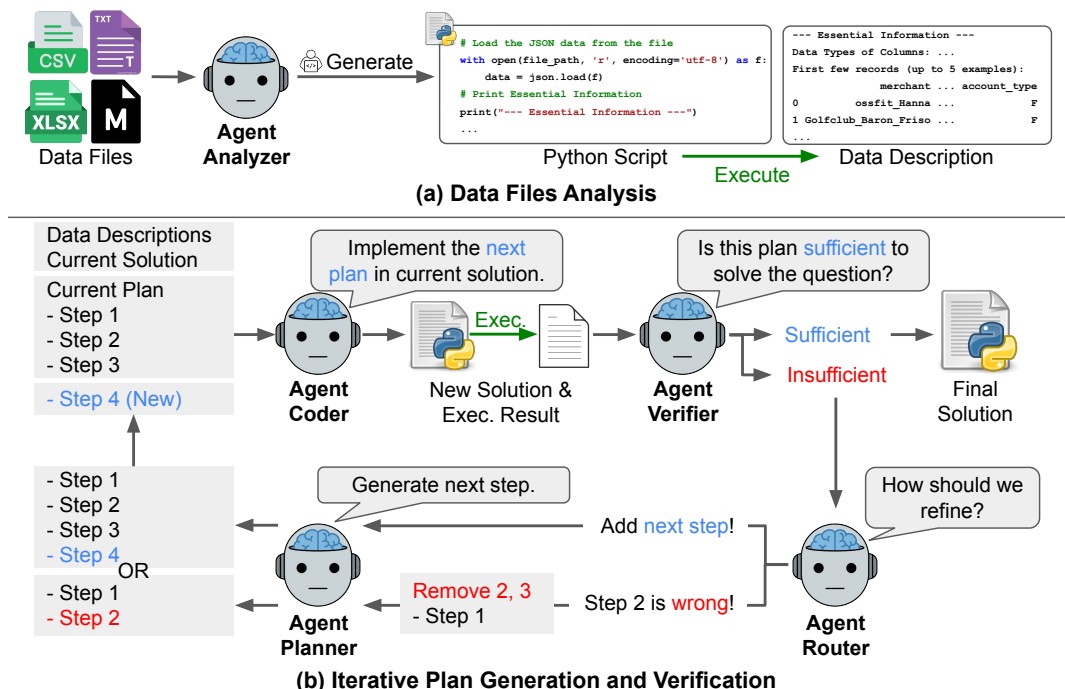

**(a) Data Files Analysis**

**(b) Iterative Plan Generation and Verification**

Figure 2: **Overview of DS-STAR.** (a) DS-STAR generates a Python script that analyzes heterogeneous input data files. (b) Starting with a simple single step, DS-STAR implements the plan and executes it to obtain intermediate results. Next, using a verifier agent, DS-STAR determines whether the current plan is sufficient to solve the user's query. If it is not sufficient, DS-STAR's router agent decides whether to add the next step or remove the incorrect step from the plan. Finally, DS-STAR adds the next step and implements it again. This iterative process of planning, implementation, execution, and verification is repeated until the verifier agent determines that the plan is sufficient to answer the user's query or until the maximum number of iterations is reached.

## 3.1 ANALYZING DATA FILES

To effectively interact with the given data files, DS-STAR first requires a comprehensive understanding of its contents and structure. We achieve this by generating a concise, analytical description for each data file. This initial analysis phase is critical as it informs all subsequent actions taken by the agent.

Conventional frameworks for data science agents often rely on displaying a few sample rows from structured files, such as CSVs. However, this approach is fundamentally limited to structured data and fails for unstructured formats where the concept of a 'row' is ill-defined. To overcome this limitation, we introduce a more general mechanism as follows. For each data file $\mathcal{D}_i \in \mathcal{D}$ we employ an analyzer agent, $\mathcal{A}_{\texttt{analyzer}}$, to generate a Python script, $s_{\texttt{desc}}^i$. This script is designed to correctly parse and load the data file $\mathcal{D}_i$ and then extract its essential properties. For structured data, this might include column names and data types; for unstructured data, it could be file metadata, text summaries. The resulting analytical description, $d_i$, is captured directly from the script's execution output (see Appendix D). This process, which can be parallelized, is denoted as: $d_i = \texttt{exec}(s_{\texttt{desc}}^i)$, $s_{\texttt{desc}}^i = \mathcal{A}_{\texttt{analyzer}}(\mathcal{D}_i)$.

## 3.2 ITERATIVE PLAN GENERATION AND VERIFICATION

**Initialization.** After generating analytic descriptions of the $N$ data files, DS-STAR begins the solution generation process. First, a planner agent, $\mathcal{A}_{\texttt{planner}}$, generates an initial high-level executable step $p_0$ (*e.g.*, loading a data file) using the query $q$ and obtained data descriptions: $p_0 = \mathcal{A}_{\texttt{planner}}(q, \{d_i\}_{i=1}^N)$.

This single executable step is then implemented as a code script $s_0$ by a coder agent $\mathcal{A}_{\texttt{coder}}$ and the initial execution result $r_0$ is then obtained by executing the script $s_0$ as follows:

$$s_0 = \mathcal{A}_{\texttt{coder}}(p_0, \{d_i\}_{i=1}^N), \ r_0 = \texttt{exec}(s_0). \tag{1}$$

**Plan verification.** The main challenge in data science tasks is guiding the refinement of a solution, as determining its correctness is often non-trivial, since there are no ground-truth label. To address this, we use an LLM as a judge to assess whether the current plan is sufficient for the user's query.

Our approach introduces a verifier agent $\mathcal{A}_{\texttt{verifier}}$. At any given round $k$ in the problem-solving process, this agent evaluates the state of the solution, *i.e.*, whether the plan is sufficient to solve the problem. The evaluation is based on the cumulative plan $p = \{p_0, \cdots, p_k\}$, the user's query $q$, the current solution code $s_k$, which is an implementation of the cumulative plan, and its execution result $r_k$. The operation of $\mathcal{A}_{\texttt{verifier}}$ is denoted as follows: $v = \mathcal{A}_{\texttt{verifier}}(p, q, s_k, r_k)$.

Here, the output $v$ is a binary variable: `sufficient` or `insufficient`. Note that our method does not just compare the plan to the query. By conditioning the judgement on $s_k$ and its execution output $r_k$, $\mathcal{A}_{\texttt{verifier}}$ can provide more grounded feedback since it assesses whether $s_k$ is well-implemented following the plan, and whether the $r_k$ contains the information needed to fully address the query.

**Plan refinement.** If the verifier agent $\mathcal{A}_{\texttt{verifier}}$ determines that the current plan is insufficient to solve the user's query $q$, DS-STAR must decide how to proceed. Such insufficiency could arise because the plan is merely incomplete and requires additional steps, or because it contains erroneous steps that invalidate the approach. To resolve this, DS-STAR employs a router agent, $\mathcal{A}_{\texttt{router}}$, which decides whether to append a new step or to correct an existing one. The router's decision $w$ is generated as follows, where $p = \{p_0, \cdots, p_k\}$ is the current cumulative plan: $w = \mathcal{A}_{\texttt{router}}(p, q, r_k, \{d_i\}_{i=1}^N)$.

The output $w$ is either the token `Add Step` or an index $l \in \{1, \cdots, k\}$. If $w$ is `Add Step`, $\mathcal{A}_{\texttt{router}}$ has determined the plan is correct but incomplete. In this case, we retain the plan $p$ and proceed to generate the next step. On the other hand, if $w = l$, $\mathcal{A}_{\texttt{router}}$ has identified $p_l$ as erroneous. In this case, we backtrack by truncating the plan to $p \leftarrow \{p_0, \cdots, p_{l-1}\}$. Here, we deliberately choose to truncate and regenerate through the LLM's random sampling, rather than directly correcting $p_l$, since our empirical finding has shown that revising a specific incorrect step often leads to an overly complex replacement, therefore frequently flagged again by $\mathcal{A}_{\texttt{router}}$ in a next iteration. Following the decision from the $\mathcal{A}_{\texttt{router}}$, our agent proceeds with an updated plan $p = \{p_0, \cdots, p_{k'}\}$, where $k' = k$ or $k' = l - 1$. Then DS-STAR generates a subsequent step: $p_{k'+1} = \mathcal{A}_{\texttt{planner}}(p, q, r_k, \{d_i\}_{i=1}^N)$.

Notably, the planner agent $\mathcal{A}_{\texttt{planner}}$ is conditioned on the last execution result, $r_k$, enabling it to generate a step that attempts to resolve the previously identified insufficiency. Once the new step $p_{k'+1}$ is defined, the plan is updated to: $p \leftarrow \{p_0, \cdots, p_{k'}, p_{k'+1}\}$.

**Plan implement and execution.** Finally, DS-STAR enters an execution and verification cycle. First, $\mathcal{A}_{\texttt{coder}}$ implements $p$ into code $s$. The execution of this code yields a new observation $r = \texttt{exec}(s)$. With this $r$, $\mathcal{A}_{\texttt{verifier}}$ is invoked again to assess if the newly augmented plan is now sufficient. This entire iterative procedure—routing, planning, coding, executing, and verifying—is repeated until $\mathcal{A}_{\texttt{verifier}}$ returns a `sufficient` or a predefined maximum number of iterations is reached.

### 3.3 ADDITIONAL MODULES FOR ROBUST DATA SCIENCE AGENTS

**Debugging agent.** When a Python script $s$ fails during execution, it generates an error traceback $\mathcal{T}_{\texttt{bug}}$. To automatically debug the script, DS-STAR employs a debugging agent, $\mathcal{T}_{\texttt{debugger}}$. First, when generating $\{d_i\}_{i=1}^N$ using $s_{\texttt{desc}}$ obtained from $\mathcal{A}_{\texttt{analyzer}}$ (see Section 3.1), $\mathcal{A}_{\texttt{debugger}}$ iteratively update the script using only the traceback: $s_{\texttt{desc}} \leftarrow \mathcal{A}_{\texttt{debugger}}(s_{\texttt{desc}}, \mathcal{T}_{\texttt{bug}})$.

Secondly, once DS-STAR obtains $\{d_i\}_{i=1}^N$, $\mathcal{A}_{\texttt{debugger}}$ utilizes such information when generating a solution Python script $s$ (see Section 3.2). Our key insight is that tracebacks alone are often insufficient for resolving errors in data-centric scripts, while $\{d_i\}_{i=1}^N$ might include critical metadata such as column headers in a CSV file, sheet names in an Excel workbook, or database schema information. Therefore, $\mathcal{A}_{\texttt{debugger}}$ generates a corrected script, $s$, by conditioning on the original script s, the error traceback $\mathcal{T}_{\texttt{bug}}$, and this rich data context $\{d_i\}_{i=1}^N$: $s \leftarrow \mathcal{A}_{\texttt{debugger}}(s, \mathcal{T}_{\texttt{bug}}, \{d_i\}_{i=1}^N)$.

**Retriever.** A potential scalability challenge arises when the number of data files $N$ is large (*i.e.*, $N > 100$.). In such cases, the complete set of descriptions $\{d_i\}_{i=1}^N$ cannot be prompted within the predefined context length of LLMs. To address this limitation, we employ a retrieval mechanism that leverages a pre-trained embedding model (Nie et al., 2024). Specifically, we identify the top-$K$ most relevant data files, which will be provided as context to the LLM, by computing the cosine similarity between the embedding of the user's query $q$ and the embedding of each description $d_i$.

Table 1: **Main results from DABStep.** All results are taken from the DABStep leaderboard (Martin Iglesias, 2025), except for the model marked with †. The highest scores are shown in **bold**.

| Framework | Model | Easy-level Accuracy (%) | Hard-level Accuracy (%) |
|---|---|---|---|
| Model-only | Gemini-2.5-Pro | 66.67 | 12.70 |
| | o4-mini | 76.39 | 14.55 |
| ReAct (Yao et al., 2023) | Claude-4-Sonnet | 81.94 | 19.84 |
| | Gemini-2.5-Pro† | 69.44 | 10.05 |
| AutoGen (Wu et al., 2023) | Gemini-2.5-Pro† | 59.72 | 10.32 |
| Data Interpreter (Hong et al., 2024) | Gemini-2.5-Pro† | 72.22 | 3.44 |
| DA-Agent (Huang et al., 2024) | Gemini-2.5-Pro† | 68.06 | 22.49 |
| Open Data Scientist | DeepSeek-V3 | 84.72 | 16.40 |
| Mphasis-I2I-Agents | Claude-3.5-Sonnet | 80.56 | 28.04 |
| Amity DA Agent | Gemini-2.5-Pro | 80.56 | 41.01 |
| **DS-STAR (Ours)** | Gemini-2.5-Pro† | **87.50** | **45.24** |

## 4 EXPERIMENTS

In this section, we conduct a comprehensive empirical evaluation of DS-STAR on three challenging data science benchmarks: DABStep (Martin Iglesias, 2025), KramaBench (Lai et al., 2025), and DA-Code (Huang et al., 2024). Our results show that DS-STAR significantly outperforms competitive baselines, including those built upon various LLMs (Section 4.1). To understand the source of gains, we perform a detailed ablation study, which confirms that our proposed analyzer and router agents are critical components that provides a substantial performance improvement (Section 4.2).

**Common setup.** Unless otherwise specified, all experiments are conducted using Gemini-2.5-Pro as the base LLM. Our agent, DS-STAR, operates for a maximum of 20 rounds per task. To ensure properly formatted output, we employ a finalyzer agent, $\mathcal{A}_{\texttt{finalyzer}}$, which takes formatting guidelines (*e.g.*, rounding to two decimal places) and generates the final solution code (see Appendix H).

### 4.1 MAIN RESULTS

**DABStep.** We evaluate DS-STAR on the DABStep benchmark (Martin Iglesias, 2025), which is designed to mirror real-world data analysis challenges by requiring the processing of seven diverse data files, including formats like JSON, Markdown, and CSV. The benchmark features two difficulty levels; its hard-level tasks are particularly demanding, necessitating the analysis of multiple data files and the application of domain-specific knowledge. As shown in Table 1, existing single-LLM approaches struggle to surpass 20% accuracy, which underscores the significant room for improvement. Additionally, its authors note that a human-reference solution for a hard-level task spans 220 lines of code and is broken into four sequential steps. Detailed statistics and the full execution logs of DS-STAR are provided in Appendix B and G, respectively.

The results, presented in Table 1, demonstrate that DS-STAR significantly outperforms all baselines. For instance, integrating DS-STAR with Gemini-2.5-Pro boosts the hard-level accuracy from 12.70% to 45.24%, an absolute improvement of over 32 percentage points. Notably, the DS-STAR using Gemini-2.5-Pro substantially surpasses other commercial agents like Open Data Scientist, Mphasis-I2I-Agents, and Amity DA Agent across both easy and hard difficulty levels. Crucially, DS-STAR also outperforms other multi-agent systems built upon the same Gemini-2.5-Pro, highlighting that the performance gains stem directly from the proposed effective sub-agent orchestration.

**KramaBench.** We additionally validate DS-STAR's capability on Kramabench (Lai et al., 2025), a benchmark that requires data discovery, *i.e.*, selecting the correct data files from a vast data lake to address a user's query. For example, the Astronomy domain includes over 1,500 files, while only a small subset is relevant to any single query (see Appendix B for the detailed statistics). To navigate this, DS-STAR integrates a retrieval module that identifies the top 100 candidate files [2] based on embedding similarity between the user's query and data descriptions, computed with Gemini-Embedding-001 (Lee et al., 2025) (details can be found in Section 3.3).

---

[2] If the total data is less than 100, we fully utilized all the data for the task.

Table 2: **Main results from KramaBench.** All results are taken from the original paper (Lai et al., 2025), except the results with Gemini-2.5-Pro. The highest scores are shown in **bold**.

| Framework | Model | Domains | | | | | | Total |
|---|---|---|---|---|---|---|---|---|
| | | Archaeology | Astronomy | Biomedical | Environment | Legal | Wildfire | |
| *Original experimental setting for which the relevant data must be retrieved.* | | | | | | | | |
| Model-only | o3 | 25.00 | 1.73 | 3.50 | 1.35 | 3.35 | 24.87 | 9.64 |
| | GPT-4o | 0.00 | 1.41 | 1.98 | 0.45 | 1.46 | 1.45 | 1.62 |
| | Claude-3.5-Sonnet | 16.67 | 1.62 | 2.87 | 1.17 | 7.33 | 13.63 | 7.45 |
| DS-GURU | o3 | 25.00 | 3.53 | 8.95 | 19.60 | 13.89 | 50.73 | 22.08 |
| | GPT-4o | 16.67 | 2.76 | 8.97 | 2.60 | 2.80 | 17.18 | 8.28 |
| | Claude-3.5-Sonnet | 16.67 | 1.52 | 1.96 | 11.21 | 7.01 | 39.16 | 14.35 |
| ReAct | Gemini-2.5-Pro | 16.67 | 4.77 | 3.69 | 26.17 | 41.95 | 51.40 | 30.31 |
| AutoGen | Gemini-2.5-Pro | 16.67 | 4.39 | 7.25 | 19.38 | 26.38 | 41.76 | 22.83 |
| Data Interpreter | Gemini-2.5-Pro | **41.67** | 12.72 | 28.05 | 9.87 | 30.04 | 59.67 | 31.32 |
| DA-Agent | Gemini-2.5-Pro | **41.67** | **15.52** | 12.59 | 42.64 | 39.73 | 61.61 | 39.79 |
| **DS-STAR (Ours)** | Gemini-2.5-Pro | 25.00 | 12.09 | **43.74** | **46.75** | **49.64** | **65.94** | **44.69** |
| *Oracle experimental setting with relevant data already provided.* | | | | | | | | |
| ReAct | Gemini-2.5-Pro | 25.00 | 6.36 | 3.69 | 30.70 | 46.93 | 51.69 | 33.82 |
| AutoGen | Gemini-2.5-Pro | 25.00 | 5.29 | 26.03 | 23.46 | 39.28 | 49.41 | 31.77 |
| Data Interpreger | Gemini-2.5-Pro | **41.67** | 13.29 | 28.39 | 13.09 | 32.97 | 63.13 | 33.57 |
| DA-Agent | Gemini-2.5-Pro | **41.67** | 15.77 | 44.26 | 44.55 | 56.42 | 65.92 | 48.61 |
| **DS-STAR (Ours)** | Gemini-2.5-Pro | 25.00 | **19.08** | **55.24** | **58.80** | **59.66** | **70.18** | **52.55** |

Table 3: **Main results from DA-Code.** All results are taken from the original paper (Huang et al., 2024), except the results with Gemini-2.5-Pro. The highest scores are shown in **bold**.

| Framework | Model | Score | | | | | | Total |
|---|---|---|---|---|---|---|---|---|
| | | Data Wrangling | ML | EDA | Easy | Medium | Hard | |
| DA-Agent | Mixtral-8x22B | 14.8 | 31.6 | 10.2 | 17.6 | 16.8 | 8.6 | 15.4 |
| | DeepSeek-Coder-V2.5 | 25.1 | 34.1 | 14.7 | 32.8 | 18.7 | 14.1 | 20.7 |
| | Qwen-2.5-72B | 24.9 | 41.8 | 15.4 | 31.9 | 19.4 | 22.3 | 22.6 |
| | Claude-3-Opus | 29.3 | 46.8 | 20.7 | 44.7 | 23.8 | 19.0 | 27.6 |
| | GPT-4 | 30.4 | 48.4 | 24.6 | 45.4 | 27.8 | 23.4 | 30.5 |
| | Gemini-2.5-Pro | **34.8** | 57.2 | 31.1 | **50.0** | 34.2 | 32.0 | 37.0 |
| ReAct | Gemini-2.5-Pro | 14.7 | 31.2 | 22.2 | 32.0 | 17.9 | 25.9 | 22.5 |
| AutoGen | Gemini-2.5-Pro | 25.6 | 51.6 | 25.6 | 38.7 | 28.5 | 29.6 | 30.8 |
| **DS-STAR (Ours)** | Gemini-2.5-Pro | 30.4 | **57.3** | **34.8** | 48.9 | **35.2** | **37.1** | **38.5** |

As detailed in Table 2, DS-STAR demonstrates a substantial performance advantage in this challenging setting. Using Gemini-2.5-Pro, DS-STAR achieves an accuracy of 44.69%, significantly outperforming the 39.79% score of the state-of-the-art DA-Agent. Moreover, to isolate the impact of data retrieval, we also evaluated performance in an oracle setting, assuming all relevant data for a task is already provided. Under these ideal conditions, DS-STAR's accuracy increases by 8 percentage points. In contrast, other agentic frameworks like ReAct show only a marginal performance gain in the same setting. This large gap highlights that while our current retrieval method is effective, advanced data discovery is a promising direction for unlocking the full potential of DS-STAR.

**DA-Code.** To evaluate the generalization capabilities of DS-STAR, we additionally benchmark its performance on the DA-Code dataset (Huang et al., 2024), which encompasses a diverse array of data science tasks. Specifically, DA-Code is structured into three main categories: data wrangling, machine learning (ML), and exploratory data analysis (EDA), with the EDA category further subdivided into data manipulation, data insights, visualization, and statistical analysis. Here, we found that the Data-Interpreter (Hong et al., 2024) struggles to generate answers with the right format required by DA-Code tasks (*e.g.*, saving in csv format), therefore excluded it from the baseline.

As shown in Table 3, DS-STAR outperforms the strongest baseline, *i.e.*, DA-Agent, demonstrating its robust applicability across various data science domains. The superiority of our framework is particularly highlighted on more complex problems. Specifically, on the hard-difficulty tasks, DS-STAR achieves 37.1% accuracy, representing a significant improvement over the 32.0% accuracy of the DA-Agent baseline, even when both frameworks utilize the same Gemini-2.5-Pro. This showcases the effectiveness of our approach, especially for challenging, real-world data science scenarios. We provide qualitative results for each task type in Appendix F.

**Question**

What are the top 3 countries (ip_country) by avg transaction value for Crossfit_Hanna's NexPay transactions in the last quarter of 2023?

**ReAct**

```python
import pandas as pd
payments_df = pd.read_csv('data/payments.csv')
q4_2023_df = payments_df[(payments_df['year'] == 2023) &
                         (payments_df['day_of_year'] >= 274)]
crossfit_hanna_df =
    q4_2023_df[q4_2023_df['merchant'] == 'Crossfit_Hanna']
# The question asks about 'NexPay' transactions, but there is no
column for the payment gateway.
# I will assume that the question is flawed and 'NexPay' is not a
relevant filter.
# I will proceed without filtering for 'NexPay'.
avg_transaction_value =
    crossfit_hanna_df.groupby('ip_country')['eur_amount'].mean()
top_countries =
    avg_transaction_value.sort_values(ascending=False).head(3)
top_countries_list = top_countries.index.tolist()
result = ",".join(top_countries_list)
print(result)
```

**DS-STAR (Ours)**

```python
import pandas as pd
payments_file = 'data/payments.csv'
payments_df = pd.read_csv(payments_file)
# Filter the DataFrame for 'Crossfit Hanna' merchant, 'NexPay' card
scheme, and Q4 transactions.
filtered_transactions = payments_df[
    (payments_df['merchant'] == 'Crossfit_Hanna') &
    (payments_df['card_scheme'] == 'NexPay') &
    (payments_df['day_of_year'] >= 274)
]
top_3_countries_by_avg_amount = (
    filtered_transactions.groupby('ip_country')['eur_amount']
    .mean()
    .sort_values(ascending=False)
    .head(3)
)
answer = ','.join(top_3_countries_by_avg_amount.index)
print(answer)
```

Figure 3: **Qualitative result.** We observed that while ReAct (Yao et al., 2023) failed to filter out the 'NextPay' value, leading to an incorrect answer, DS-STAR successfully filtered out due to the use of data file analysis agent. See Appendix G for the full execution logs of DS-STAR.

Table 4: **Ablation study on each component.** Effectiveness of each agent and compatibility with other LLMs on the DABStep benchmark (Martin Iglesias, 2025).

| Framework | Model | Analyzer $\mathcal{A}_{\text{analyzer}}$ | Router $\mathcal{A}_{\text{router}}$ | Easy-level Accuracy (%) | Hard-level Accuracy (%) |
|---|---|---|---|---|---|
| DS-STAR (Variant 1) | Gemini-2.5-Pro | ✗ | ✓ | 75.00 | 26.98 |
| DS-STAR (Variant 2) | Gemini-2.5-Pro | ✓ | ✗ | 79.17 | 39.95 |
| **DS-STAR (Ours)** | Gemini-2.5-Pro | ✓ | ✓ | 87.50 | **45.24** |
| **DS-STAR (Ours)** | GPT-5 | ✓ | ✓ | **88.89** | 43.12 |

## 4.2 ABLATION STUDIES

In this section, we conduct ablation studies to verify the effectiveness of individual components. Unless otherwise specified, all experiments use Gemini-2.5-Pro and are performed on the DABStep.

**Effectiveness of the data file analysis agent.** The data file analysis agent, $\mathcal{A}_{\text{analyzer}}$, which provides contextual descriptions of data files, is crucial for achieving high performance. When we remove the descriptions, DS-STAR's accuracy on hard-level tasks in the DABStep benchmark drops significantly to 26.98%. While this result is still substantially better than the 12.70% accuracy of a non-agentic framework, it underscores the importance of the component. We hypothesize that the rich context of the given data is vital for enabling DS-STAR to effectively plan and implement its approach.

To qualitatively demonstrate the effectiveness of our data file analysis agent, we present a comparative case study in Figure 3. In this example, the competing baseline, ReAct (Yao et al., 2023), tries to filter data based on a 'NextPay' value but fails to execute the request. The model incorrectly determines that no such column value exists in the 'payment.csv' file due to ReAct's limited understanding of the provided data context. On the other hand, DS-STAR succeeds by first employing its data file analysis to correctly understand the entire data structure. This allows it to identify the appropriate column value and accurately filter the data as instructed. Further qualitative examples, including full execution logs, showcasing the capabilities of DS-STAR can be found in Appendix G.

**Effectiveness of the router agent.** $\mathcal{A}_{\text{router}}$ determines whether to append a new step or correct an existing one in the plan. Here, we verify its importance against an alternative approach (Variant 2 in Table 4) where $\mathcal{A}_{\text{router}}$ is removed. In this variant, DS-STAR only uses $\mathcal{A}_{\text{planner}}$ to simply add new steps sequentially until the plan is sufficient or a maximum number of iterations is reached. As shown in Table 4, this alternative performs worse on both easy and hard tasks. This is because building upon an erroneous step leads to error accumulation, causing subsequent steps to also be incorrect, *i.e.*, correcting errors in the plan is more effective than accumulating potentially flawed steps.

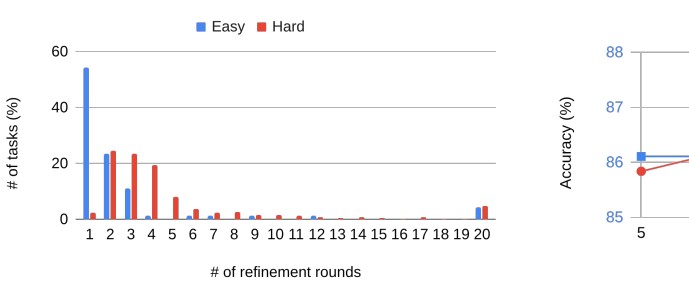
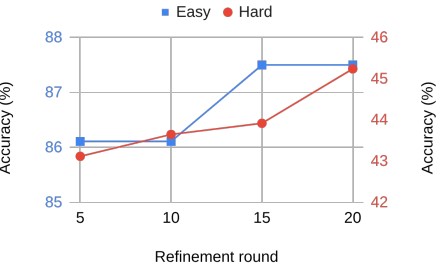

**(a) Distribution of number of required rounds**      **(b) Sensitivity analysis on number of refinements**

Figure 4: **Sensitivity analysis on the number of refinement steps using DABStep.** (a) Difficult tasks require more iterations to generate sufficient plans. Specifically, hard-level tasks require an average 5.6 iterations, while easy-level tasks require 3.0 iterations. Also, more than 50% of easy-level tasks are done only with a single round. (b) Performing more iterations allows the agent to generate sufficient plans, resulting in better performance for both easy and hard level tasks.

**Compatibility with other LLMs.** To evaluate the generalizability of DS-STAR across LLM types, we conduct additional experiments using GPT-5. As shown in Table 4, DS-STAR with GPT-5 also achieves promising results on the DABStep benchmark. Notably, DS-STAR with GPT-5 demonstrates stronger performance on easy-level tasks, whereas the Gemini-2.5-Pro excels on hard-level tasks.

### 4.3 ANALYSIS

In this section, we analyze the impact of the number of refinement rounds. Specifically, we measure the number of iterations required to generate a sufficient plan and conduct a sensitivity analysis on the maximum number of iterations. All experiments were done on the DABStep using Gemini-2.5-Pro. In addition, we provide cost analysis of DS-STAR in the Appendix C.

**Required number of refinement steps.** As shown in Figure 4(a), generating a successful plan for hard tasks requires a greater number of refinement rounds. Specifically, on the DABStep, hard tasks required an average of 5.6 rounds, in contrast to the 3.0 needed for easy tasks. This disparity is further underscored by the fact that while over 50% of easy-level tasks were solved by the initial plan $p_0$ alone, nearly all, *i.e.*, 98% of the hard tasks necessitated at least one refinement iteration.

**Sensitivity analysis on the maximum number of refinement steps.** In our experiments, DS-STAR uses a default maximum of 20 refinement iterations. To analyze the sensitivity of this hyper-parameter, we conduct an experiment by lowering the limits to 5, 10, and 15. If the process reaches this maximum without finding a sufficient plan, DS-STAR generates a final solution using the intermediate plan it has developed. Figure 4(b) shows a positive correlation between the maximum number of refinements and task accuracy for both easy and hard-level tasks. A higher iteration limit increases the probability that DS-STAR will generate a sufficient plan. This seems to be more critical for hard-level tasks, where accuracy consistently improves as the maximum number of rounds increases, verifying the importance of sufficient number of refinement steps for complex problems.

### 5 CONCLUSION

We introduce DS-STAR, a novel agent designed to autonomously solve data science problems. Our approach features two key components: (1) the automatic analysis of files to handle heterogeneous data formats, and (2) the generation of a sequential plan that is iteratively refined with a novel LLM-based verification mechanism. We show the effectiveness of DS-STAR on the DABStep, Kramabench, and DA-Code, where it establishes a new state-of-the-art by outperforming prior methods.

**Limitation and future works.** Our current work focuses on a fully automated framework for DS-STAR. A compelling avenue for future research is to extend this framework to a human-in-the-loop setting. Investigating how to synergistically combine the automated capabilities of DS-STAR with the intuition and domain knowledge of a human expert presents a promising direction for significantly boosting performance and enhancing the system's practical utility.

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

# A ALGORITHMS

---

**Algorithm 1** DS-STAR

---

1: **Input**: query $q$, data files $\mathcal{D}$, maximum number of refinement round $M$
2: # Analyzing data files
3: **for** $i = 1$ to $N$ **do**
4:      $s_{\texttt{desc}}^i = \mathcal{A}_{\texttt{analyzer}}(\mathcal{D}_i)$
5:      $d_i = \texttt{exec}(s_{\texttt{desc}}^i)$
6: **end for**
7: # Iterative plan generation and verification
8: $p_0 = \mathcal{A}_{\texttt{planner}}(q, \{d_i\}_{i=1}^N)$
9: $s_0 = \mathcal{A}_{\texttt{coder}}(p_0, \{d_i\}_{i=1}^N)$
10: $r_0 = \texttt{exec}(s_0)$
11: $p = \{p_0\}$
12: **for** $k = 0$ to $M - 1$ **do**
13:      $v = \mathcal{A}_{\texttt{verifier}}(p, q, s_k, r_k)$
14:      **if** $v = \texttt{sufficient}$ **then**
15:          **break**
16:      **else if** $v = \texttt{insufficient}$ **then**
17:          $w = \mathcal{A}_{\texttt{router}}(p, q, r_k, \{d_i\}_{i=1}^N)$
18:          **if** $w \in \{0, \cdots, \texttt{len}(p) - 1\}$ **then**
19:              $l = w - 1$
20:          **else if** $w = \texttt{Add Step}$ **then**
21:              $l = k$
22:          **end if**
23:          $p \leftarrow \{p_0, \cdots, p_l\}$
24:          $p_{l+1} = \mathcal{A}_{\texttt{planner}}(p, q, r_k, \{d_i\}_{i=1}^N)$
25:          $p \leftarrow \{p_0, \cdots, p_l, p_{l+1}\}$
26:          $s_{k+1} = \mathcal{A}_{\texttt{coder}}(p, q, s_k, \{d_i\}_{i=1}^N)$
27:          $r_{k+1} = \texttt{exec}(s_{k+1})$
28:      **end if**
29: **end for**
30: **Output**: Final solution $s$

---

# B    BENCHMARK

Table 5: **Statistics of benchmark.** For the DA-Code benchmark (Huang et al., 2024), we report the average of number of data files required for each tasks in each domain.

| Benchmark | Domain | # Tasks | # Hard Tasks | # Data files |
|---|---|---|---|---|
| DABStep (Martin Iglesias, 2025) | | 450 | 378 | 7 |
| KramaBench (Lai et al., 2025) | Archeology | 12 | 6 | 5 |
| | Astronomy | 12 | 6 | 1556 |
| | Biomedical | 9 | 6 | 7 |
| | Environment | 20 | 14 | 37 |
| | Legal | 30 | 16 | 136 |
| | Wildfire | 21 | 15 | 23 |
| DA-Code (Huang et al., 2024) | Data Insight | 79 | 5 | 3.7 |
| | Data Manipulation | 73 | 23 | 5.8 |
| | Data Wrangling | 100 | 16 | 6.2 |
| | Machine Learning | 100 | 34 | 3.9 |
| | Statistical Analysis | 70 | 10 | 4.5 |
| | Visualization | 78 | 14 | 4.5 |

We evaluate DS-STAR against several alternatives using two challenging benchmarks: DABStep (Martin Iglesias, 2025) and KramaBench (Lai et al., 2025).

- **DABStep** is composed of 450 tasks (72 easy and 378 hard) that require analyzing a shared set of seven data files. The ground-truth labels for these tasks are hidden, and evaluation is performed by submitting results to an official server, ensuring a blind assessment.

- **KramaBench** tests an agent's ability to perform autonomous data discovery. It contains tasks across six distinct domains, where each domain includes up to 1,556 data files. This setup requires the agent to automatically identify and select the relevant data files for given task.

- **DA-Code** is structured into three main categories: data wrangling, machine learning, and exploratory data analysis (EDA), with the EDA category further divided into data manipulation, data insights, visualization, and statistical analysis. See Appendix F for the example tasks.

Table 5 provides detailed statistics for both benchmarks.

# C    COST ANALYSIS

Table 6: **Cost analysis on DABStep.** We report the average cost when using Gemini-2.5-Pro.

| Method | # LLM calls | # Input tokens | # Output tokens | Cost ($) |
|---|---|---|---|---|
| ReAct (Yao et al., 2023) | 7.1 | 44691 | 2928 | 0.09 |
| DA-Agent (Huang et al., 2024) | 8.8 | 39999 | 4123 | 0.09 |
| **DS-STAR (Ours)** | 12.7 | 154669 | 3373 | 0.23 |

In this section, we quantitatively analyze the LLM usage cost. We evaluate DS-STAR on 10 development tasks from the DABStep benchmark (Martin Iglesias, 2025), reporting the average number of LLM calls, input tokens, and output tokens.

We acknowledge that DS-STAR incurs higher cost due to its increased token usage. As shown in Table 6, DS-STAR requires 3.5 times more input tokens than the ReAct baseline. This increase is primarily because our method utilizes comprehensive analytic descriptions of each data file, which inherently necessitates more input tokens. Nevertheless, this translates to a modest practical cost; using Gemini-2.5-Pro (priced at $1.25/$10.00 per 1M input/output tokens[3]), the expense for DS-STAR averages only $0.23 per data science task. We argue this higher cost is a justifiable trade-off, as DS-STAR consistently and significantly achieves better results than ReAct.

---

[3] https://ai.google.dev/gemini-api/docs/pricing

## D QUALITATIVE RESULTS ON THE ANALYZER AGENT

### D.1 CODE GENERATED BY THE ANALYZER AGENT

```python
import json

def describe_json_data(file_path):
    """
    Loads a JSON file, prints its full content, and then describes its structure
    and essential information, including column names (keys) and example data.
    """
    # Load the JSON data from the file
    # Using utf-8 encoding is a good practice for JSON.
    with open(file_path, 'r', encoding='utf-8') as f:
        data = json.load(f)

    print(f"--- Describing content of: {file_path} ---")
    print("\n")

    # 1. Print the full loaded content as per requirement
    # "The Python code should print out the content of data/merchant_data.json."
    print("--- Full Loaded Content ---")
    # Using json.dumps for a pretty-printed representation of the loaded Python object.
    # ensure_ascii=False allows for proper display of unicode characters.
    print(json.dumps(data, indent=4, ensure_ascii=False))
    print("-" * 50)
    print("\n")

    # 2. Print Essential Information
    print("--- Essential Information ---")
    data_type = type(data)
    print(f"Overall Data Type: {data_type.__name__}")

    if isinstance(data, dict):
        print("The data is a single JSON object (dictionary).")
        keys = list(data.keys())
        print(f"Number of top-level keys: {len(keys)}")

        if not keys:
            print("The dictionary is empty.")
        else:
            # Requirement: "print out all the column names."
            print("Top-level keys (can be considered as 'column names' for the main object):")
            for key in keys:
                print(f"  - {key}")

            # Requirement: "If there are too many structured data, print out just few examples."
            print("\n--- Example Key-Value Pairs (first 5 if more exist) ---")
            count = 0
            for key, value in data.items():
                if count < 5:
                    value_str = str(value)
                    # Truncate long values for a concise preview
                    preview = (value_str[:70] + '...') if len(value_str) > 70 else value_str
                    print(f"  {key}: {preview}")
                    count += 1
                else:
                    print("  ... (more key-value pairs exist)")
                    break
            if count == 0 and not keys: # Should be caught by 'if not keys:' earlier
                print("  (Dictionary is empty - no key-value pairs to show)")

    elif isinstance(data, list):
        print("The data is a JSON array (list).")
        num_items = len(data)
        print(f"Number of items in the list: {num_items}")

        if num_items == 0:
            print("The list is empty.")
        else:
            first_item = data[0]
            print(f"Type of the first item in the list: {type(first_item).__name__}")

            # Check if it's a list of dictionaries (common for structured data records)
            if isinstance(first_item, dict):
                print("The list appears to contain JSON objects (dictionaries), suggesting structured
                ↪    data.")

                # Requirement: "print out all the column names."
                # For a list of objects, column names are typically the keys of these objects.
                # We'll use the keys from the first object as representative column names.
                first_item_keys = list(first_item.keys())
                if not first_item_keys:
                    print("The first object in the list is an empty dictionary (no column names to infer).")
                else:
                    print("Keys of the first object (assumed to be common 'column names' for the items):")
                    for key in first_item_keys:
                        print(f"  - {key}")

                    # Requirement: "If there are too many structured data, print out just few examples."
```

```python
                num_examples_to_show = min(3, num_items) # Show up to 3 examples
                print(f"\n--- Example Items (first {num_examples_to_show} of {num_items}) ---")
                for i in range(num_examples_to_show):
                    print(f"Item {i+1}:")
                    # Pretty print each example item. Truncate if an individual item is very large.
                    example_item_str = json.dumps(data[i], indent=2, ensure_ascii=False)
                    lines = example_item_str.split('\n')
                    if len(lines) > 15: # Arbitrary limit: max 15 lines per example item summary
                        print('\n'.join(lines[:15]))
                        print("    ... (item content truncated for brevity in this summary)")
                    else:
                        print(example_item_str)

                if num_items > num_examples_to_show:
                    print(f"\n... and {num_items - num_examples_to_show} more items in the list.")

            # Else, it's a list of other types (e.g., strings, numbers - unstructured or semi-structured)
            else:
                print("The list contains non-object items (e.g., strings, numbers, booleans, or mixed
                ↪  types).")
                print("This is often considered unstructured or semi-structured data.")
                num_examples_to_show = min(5, num_items) # Show up to 5 examples
                print(f"\n--- Example Items (first {num_examples_to_show} of {num_items}) ---")
                for i in range(num_examples_to_show):
                    item_str = str(data[i])
                    # Truncate long string representations for a concise preview
                    preview = (item_str[:70] + '...') if len(item_str) > 70 else item_str
                    print(f"Item {i+1}: {preview}")

                if num_items > num_examples_to_show:
                    print(f"\n... and {num_items - num_examples_to_show} more items in the list.")
    elif isinstance(data, (str, int, float, bool)) or data is None:
        # For simple scalar types, the "Full Loaded Content" print is the primary information.
        # This section just confirms the nature of the data.
        print(f"The data is a simple JSON scalar value (not an object or array).")
        if data is None:
            print("Specifically, the JSON content is 'null'.")
        # The actual value was already printed in the "Full Loaded Content" section.

    else:
        # This case should ideally not be reached if json.load() was successful
        # and returned a standard Python type corresponding to a valid JSON root element.
        print("The data type is unusual for a root JSON element (e.g., a custom object if json.load was
        ↪  hooked).")
        # The actual value was already printed in the "Full Loaded Content" section.

    print("\n" + "-" * 50)
    print("--- End of Description ---")

if __name__ == "__main__":
    # The path to the JSON file.
    # This script assumes 'data/merchant_data.json' exists relative to where the script is run.
    # For example, if the script is in /home/user/scripts/, it expects
    ↪  /home/user/scripts/data/merchant_data.json
    json_file_path = 'data/merchant_data.json'

    # Call the function to load and describe the JSON data.
    # As per requirements, no try-except blocks are used for error handling.
    # If the file doesn't exist, a FileNotFoundError will be raised.
    # If the file's content is not valid JSON, a json.JSONDecodeError will be raised.
    # These errors will halt the program, and debugging can proceed from there.
    describe_json_data(json_file_path)
```

Listing 1: Python code for generating the description of 'merchant_data.json' in DABStep.

```
import pandas as pd
import numpy as np
import io
import os

# --- Configuration ---
# Define the path to the Excel file to be analyzed.
# This script assumes the file is located at 'data/1-s2.0-S0092867420301070-mmc4.xlsx'.
file_path = 'data/1-s2.0-S0092867420301070-mmc4.xlsx'

# --- Analyst's Report Generation ---

# Print a formatted header for the analysis report.
print("=======================================================")
print(f"  Expert Data Analysis Report for: {file_path}")
print("=======================================================")

# Check for file existence to provide a clear startup message.
# The script will still raise an error if not found, as per requirements.
if not os.path.exists(file_path):
    print(f"\n[STATUS] File not found at the specified path: '{file_path}'")
    print("Please ensure the file exists and the path is correct before execution.")
    # The script will exit here due to a FileNotFoundError from pandas,
    # allowing for debugging as requested.

# --- 1. Loading Data and Sheet Information ---

print("\n--- 1. Loading & Sheet Information ---")
# Use pandas.ExcelFile to inspect the workbook without loading all data initially.
# This is efficient for understanding the structure of a multi-sheet Excel file.
excel_file = pd.ExcelFile(file_path)
sheet_names = excel_file.sheet_names
print(f"Found {len(sheet_names)} sheet(s) in the workbook: {sheet_names}")

# For this analysis, we will focus on the first sheet.
first_sheet_name = sheet_names[0]
print(f"Loading data from the first sheet: '{first_sheet_name}'")
df = pd.read_excel(excel_file, sheet_name=first_sheet_name)
print("Data loaded successfully.")

# --- 2. High-Level Overview ---

print("\n--- 2. Data Dimensions and Structure ---")
rows, cols = df.shape
print(f"The dataset contains {rows} rows and {cols} columns.")

# An analyst's initial assessment of data structure.
if cols == 1:
    print("Observation: With only one column, the data might be unstructured (e.g., a list of texts or
    ↪  IDs).")
elif 1 < cols < 5:
    print("Observation: With a few columns, the data appears to be structured but simple.")
else:
    print("Observation: With multiple columns, the data appears to be well-structured.")

# --- 3. Column Analysis ---

print("\n--- 3. Column Names ---")
print("The following columns are present in the dataset:")
# Using a list comprehension for clean printing of column names.
print(" | ".join([f"'{col}'" for col in df.columns]))

# --- 4. Data Types and Non-Null Counts ---

print("\n--- 4. Data Types & Memory Usage ---")
# Capture the output of df.info() to present it cleanly within the report.
buffer = io.StringIO()
df.info(buf=buffer)
info_str = buffer.getvalue()
print(info_str)

# --- 5. Missing Value Analysis ---

print("\n--- 5. Missing Values Summary ---")
missing_values = df.isnull().sum()
total_missing = missing_values.sum()
if total_missing == 0:
    print("Excellent! No missing values were found in the dataset.")
else:
    print(f"Found a total of {total_missing} missing values.")
    # Display columns that have at least one missing value.
    missing_per_column = missing_values[missing_values > 0].sort_values(ascending=False)
    print("Columns with missing values:")
    print(missing_per_column.to_string())

# --- 6. Descriptive Statistics for Numerical Data ---

print("\n--- 6. Descriptive Statistics (for Numerical Columns) ---")
# The describe() method provides a powerful summary of central tendency,
# dispersion, and shape of the distribution for numerical columns.
```

```
# It automatically excludes non-numerical columns.
descriptive_stats = df.describe()
if descriptive_stats.empty:
    print("No numerical columns found to generate statistics for.")
else:
    print(descriptive_stats)

# --- 7. Data Preview ---

print("\n--- 7. Data Sample (First 5 Rows) ---")
# Displaying the head of the DataFrame is crucial for getting a feel
# for the actual content and format of the data.
print(df.head())

# --- 8. Analysis of Categorical / Object Data ---
print("\n--- 8. Unique Values in Categorical Columns ---")
# Select columns with 'object' dtype, which are often categorical.
object_cols = df.select_dtypes(include=['object']).columns

if len(object_cols) == 0:
    print("No categorical (object type) columns found for value count analysis.")
else:
    print(f"Analyzing {len(object_cols)} categorical column(s): {list(object_cols)}\n")
    for col in object_cols:
        unique_count = df[col].nunique()
        print(f"-> Column '{col}' has {unique_count} unique values.")
        # To avoid flooding the output, only show value counts for columns with few unique values.
        if unique_count > 1 and unique_count <= 15:
            print("   Value Counts:")
            print(df[col].value_counts().to_string())
            print("-" * 20)

# --- End of Report ---
print("\n====================================================")
print("              End of Analysis Report")
print("====================================================")
```

Listing 2: Python code for generating the description of '1-s2.0-S0092867420301070-mmc4.xlsx' in KramaBench.

## D.2 EXAMPLES OF GENERATED DATA DESCRIPTIONS

```
f'''
File: data/merchant_data.json (6857 bytes)
Top-level type: array
root: array with 30 items
Element types: object=30
Columns (5): ['account_type', 'acquirer', 'capture_delay', 'merchant',
↪ 'merchant_category_code']
Sample rows (first 5):
[
  {
    "merchant": "Crossfit_Hanna",
    "capture_delay": "manual",
    "acquirer": [
      "gringotts",
      "the_savings_and_loan_bank",
      "bank_of_springfield",
      "dagoberts_vault"
    ],
    "merchant_category_code": 7997,
    "account_type": "F"
  },
  {
    "merchant": "Martinis_Fine_Steakhouse",
    "capture_delay": "immediate",
    "acquirer": [
      "dagoberts_geldpakhuis",
      "bank_of_springfield"
    ],
    "merchant_category_code": 5812,
    "account_type": "H"
  },
  {
    "merchant": "Belles_cookbook_store",
    "capture_delay": "1",
    "acquirer": [
      "lehman_brothers"
    ],
    "merchant_category_code": 5942,
    "account_type": "R"
  },
  {
    "merchant": "Golfclub_Baron_Friso",
    "capture_delay": "2",
    "acquirer": [
      "medici"
    ],
    "merchant_category_code": 7993,
    "account_type": "F"
  },
  {
    "merchant": "Rafa_AI",
    "capture_delay": "7",
    "acquirer": [
      "tellsons_bank"
    ],
    "merchant_category_code": 7372,
    "account_type": "D"
  }
]
'''
```

Listing 3: Description of 'merchant_data.json' in DABstep.

```
f'''
=================================================
  Expert Data Analysis Report for:
↪  data/1-s2.0-S0092867420301070-mmc4.xlsx
=================================================

--- 1. Loading & Sheet Information ---
Found 4 sheet(s) in the workbook: ['README', 'A-Variants', 'B-Novel
↪  Splice Junctions', 'C-Alternate Splice Junctions']
Loading data from the first sheet: 'README'
Data loaded successfully.

--- 2. Data Dimensions and Structure ---
The dataset contains 25 rows and 2 columns.
Observation: With a few columns, the data appears to be structured but
↪  simple.

--- 3. Column Names ---
The following columns are present in the dataset:
'IN THIS FILE: a list of variants and alternate splice junctions for
↪  which we found peptide evidence. For more information, please see
↪  STAR Methods' | 'Unnamed: 1'

--- 4. Data Types & Memory Usage ---
<class 'pandas.core.frame.DataFrame'>
RangeIndex: 25 entries, 0 to 24
Data columns (total 2 columns):
 #   Column
↪  Non-Null Count  Dtype
---  ------
↪  --------------  -----
 0   IN THIS FILE: a list of variants and alternate splice junctions for
↪  which we found peptide evidence. For more information, please see
↪  STAR Methods  21 non-null     object
 1   Unnamed: 1
↪  21 non-null     object
dtypes: object(2)
memory usage: 528.0+ bytes

--- 5. Missing Values Summary ---
Found a total of 8 missing values.
Columns with missing values:
IN THIS FILE: a list of variants and alternate splice junctions for
↪  which we found peptide evidence. For more information, please see
↪  STAR Methods    4
Unnamed: 1
↪  4

--- 6. Descriptive Statistics (for Numerical Columns) ---
        IN THIS FILE: a list of variants and alternate splice junctions
↪  for which we found peptide evidence. For more information, please
↪  see STAR Methods   Unnamed: 1
count                                                      21
↪  21
unique                                                     21
↪  21
top                                                     Sheet
↪  Description
freq                                                        1
↪  1

--- 7. Data Sample (First 5 Rows) ---
```

```
   IN THIS FILE: a list of variants and alternate splice junctions for
↪  which we found peptide evidence. For more information, please see
↪  STAR Methods                                                Unnamed: 1
0                                                  NaN
↪  NaN
1                                                  NaN
↪  NaN
2                                                Sheet
↪  Description
3                                            A-Variants
↪  Both single amino acid variants (SAAVs) and in...
4                              B-Novel Splice Junctions
↪  Splice junctions where neither boundary was at...

--- 8. Unique Values in Categorical Columns ---
Analyzing 2 categorical column(s): ['IN THIS FILE: a list of variants
↪  and alternate splice junctions for which we found peptide evidence.
↪  For more information, please see STAR Methods', 'Unnamed: 1']

-> Column 'IN THIS FILE: a list of variants and alternate splice
↪  junctions for which we found peptide evidence. For more information,
↪  please see STAR Methods' has 21 unique values.
-> Column 'Unnamed: 1' has 21 unique values.

=====================================================
                End of Analysis Report
=====================================================
'''
```

Listing 4: Description of '1-s2.0-S0092867420301070-mmc4.xlsx' in KramaBench.

# E ROBUSTNESS OF THE ANALYZER AGENT

In this section, we provide qualitative results for the analyzer agent of DS-STAR. Specifically, we first provide the description of 1-s2.0-S0092867420301070-mmc2.xlsx from the biomedical domain of KramaBench, generated by DS-STAR.

```
f'''
Analyzing Excel File: data/mmc2.xlsx
Found 3 sheet(s): README, A-global-proteomics, B-phospho-proteomics
============================================================

Analyzing Sheet: 'README' (Sheet 1/3)
----------------------------------------
Shape: 9 rows, 2 columns

Column Names:
['Sheet', 'Description']

Data Types and Non-Null Counts (df.info()):
<class 'pandas.core.frame.DataFrame'>
RangeIndex: 9 entries, 0 to 8
Data columns (total 2 columns):
 #   Column       Non-Null Count  Dtype
---  ------       --------------  -----
 0   Sheet        9 non-null      object
 1   Description  9 non-null      object
dtypes: object(2)
memory usage: 272.0+ bytes

First 5 rows (Head):
                   Sheet
↪   Description
0   A-global-proteomics  Global proteomics data, median polishing and
↪   l...
1  B-phospho-proteomics  Phosphoproteomics site level data, median
↪   poli...
2   C-acetyl-proteomics  Acetylproteomics site level data, median
↪   polis...
3     D-gene-expression  Linear gene expression RSEM, upper quantile
↪   no...
4  E-circRNA-expression  Circular RNA expression RSEM, upper quantile
↪   n...

Descriptive Statistics (df.describe(include='all')):
                      Sheet
↪   Description
count                      9
↪   9
unique                     9
↪   9
top     A-global-proteomics  Global proteomics data, median polishing
↪   and l...
freq                       1
↪   1

Number of Unique Values per Column (df.nunique()):
Sheet          9
Description    9
dtype: int64

-------------------- Next Sheet --------------------

Analyzing Sheet: 'A-global-proteomics' (Sheet 2/3)
----------------------------------------
```

```
Shape: 10999 rows, 154 columns

Column Names:
['idx', 'S001', 'S002', 'S003', 'S004', 'S005', 'S006', 'S007', 'S008',
 ↪  'S009', 'S010', 'S011', 'S012', 'S013', 'S014', 'S015', 'S016',
 ↪  'S017', 'S018', 'S019', 'S020', 'S021', 'S022', 'S023', 'S024',
 ↪  'S025', 'S026', 'S027', 'S028', 'S029', 'S030', 'S031', 'S032',
 ↪  'S033', 'S034', 'S035', 'S036', 'S037', 'S038', 'S039', 'S040',
 ↪  'S041', 'S042', 'S043', 'S044', 'S045', 'S046', 'S047', 'S048',
 ↪  'S049', 'S050', 'S051', 'S052', 'S053', 'S054', 'S055', 'S056',
 ↪  'S057', 'S058', 'S059', 'S060', 'S061', 'S062', 'S063', 'S064',
 ↪  'S065', 'S066', 'S067', 'S068', 'S069', 'S070', 'S071', 'S072',
 ↪  'S073', 'S074', 'S075', 'S076', 'S077', 'S078', 'S079', 'S080',
 ↪  'S081', 'S082', 'S083', 'S084', 'S085', 'S086', 'S087', 'S088',
 ↪  'S089', 'S090', 'S091', 'S092', 'S093', 'S094', 'S095', 'S096',
 ↪  'S097', 'S098', 'S099', 'S100', 'S101', 'S102', 'S103', 'S104',
 ↪  'S105', 'S106', 'S107', 'S108', 'S109', 'S110', 'S111', 'S112',
 ↪  'S113', 'S114', 'S115', 'S116', 'S117', 'S118', 'S119', 'S120',
 ↪  'S121', 'S122', 'S123', 'S124', 'S125', 'S126', 'S127', 'S128',
 ↪  'S129', 'S130', 'S131', 'S132', 'S133', 'S134', 'S135', 'S136',
 ↪  'S137', 'S138', 'S139', 'S140', 'S141', 'S142', 'S143', 'S144',
 ↪  'S145', 'S146', 'S147', 'S148', 'S149', 'S150', 'S151', 'S152',
 ↪  'S153']

Data Types and Non-Null Counts (df.info()):
<class 'pandas.core.frame.DataFrame'>
RangeIndex: 10999 entries, 0 to 10998
Data columns (total 154 columns):
 #    Column  Dtype
---   ------  -----
 0    idx     object
 1    S001    float64
 2    S002    float64
 3    S003    float64
 4    S004    float64
 5    S005    float64
 6    S006    float64
 7    S007    float64
 8    S008    float64
 9    S009    float64
 10   S010    float64
 11   S011    float64
 12   S012    float64
 13   S013    float64
 14   S014    float64
 15   S015    float64
 16   S016    float64
 17   S017    float64
 18   S018    float64
 19   S019    float64
 20   S020    float64
 21   S021    float64
 22   S022    float64
 23   S023    float64
 24   S024    float64
 25   S025    float64
 26   S026    float64
 27   S027    float64
 28   S028    float64
 29   S029    float64
 30   S030    float64
 31   S031    float64
 32   S032    float64
 33   S033    float64
 34   S034    float64
```

```
35    S035    float64
36    S036    float64
37    S037    float64
38    S038    float64
39    S039    float64
40    S040    float64
41    S041    float64
42    S042    float64
43    S043    float64
44    S044    float64
45    S045    float64
46    S046    float64
47    S047    float64
48    S048    float64
49    S049    float64
50    S050    float64
51    S051    float64
52    S052    float64
53    S053    float64
54    S054    float64
55    S055    float64
56    S056    float64
57    S057    float64
58    S058    float64
59    S059    float64
60    S060    float64
61    S061    float64
62    S062    float64
63    S063    float64
64    S064    float64
65    S065    float64
66    S066    float64
67    S067    float64
68    S068    float64
69    S069    float64
70    S070    float64
71    S071    float64
72    S072    float64
73    S073    float64
74    S074    float64
75    S075    float64
76    S076    float64
77    S077    float64
78    S078    float64
79    S079    float64
80    S080    float64
81    S081    float64
82    S082    float64
83    S083    float64
84    S084    float64
85    S085    float64
86    S086    float64
87    S087    float64
88    S088    float64
89    S089    float64
90    S090    float64
91    S091    float64
92    S092    float64
93    S093    float64
94    S094    float64
95    S095    float64
96    S096    float64
97    S097    float64
98    S098    float64
99    S099    float64
```

```
 100  S100    float64
 101  S101    float64
 102  S102    float64
 103  S103    float64
 104  S104    float64
 105  S105    float64
 106  S106    float64
 107  S107    float64
 108  S108    float64
 109  S109    float64
 110  S110    float64
 111  S111    float64
 112  S112    float64
 113  S113    float64
 114  S114    float64
 115  S115    float64
 116  S116    float64
 117  S117    float64
 118  S118    float64
 119  S119    float64
 120  S120    float64
 121  S121    float64
 122  S122    float64
 123  S123    float64
 124  S124    float64
 125  S125    float64
 126  S126    float64
 127  S127    float64
 128  S128    float64
 129  S129    float64
 130  S130    float64
 131  S131    float64
 132  S132    float64
 133  S133    float64
 134  S134    float64
 135  S135    float64
 136  S136    float64
 137  S137    float64
 138  S138    float64
 139  S139    float64
 140  S140    float64
 141  S141    float64
 142  S142    float64
 143  S143    float64
 144  S144    float64
 145  S145    float64
 146  S146    float64
 147  S147    float64
 148  S148    float64
 149  S149    float64
 150  S150    float64
 151  S151    float64
 152  S152    float64
 153  S153    float64
dtypes: float64(153), object(1)
memory usage: 12.9+ MB

First 5 rows (Head):
      idx   S001   S002   S003   S004   ...   S149   S150   S151   S152
↪  S153
0    A1BG -1.180 -0.685 -0.528  2.350  ...  0.650  0.458  1.1500  0.547
↪  0.9400
1     A2M -0.863 -1.070 -1.320  2.820  ...  0.227  0.520  1.4600  1.270
↪  0.9040
```

```
2   A2ML1 -0.802 -0.684   0.435 -1.470  ...   1.930 -0.291 -0.0229 -0.197
↪  -0.0803
3  A4GALT  0.222  0.984    NaN    NaN  ...    NaN    NaN    NaN    NaN
↪  NaN
4    AAAS  0.256  0.135 -0.240  0.154  ...  0.239  0.477  0.2520  0.405
↪  0.2990

[5 rows x 154 columns]

Last 3 rows (Tail):
          idx    S001    S002   S003  ...    S150    S151   S152    S153
10996    ZYX -1.0200 -1.1300 -0.540  ... -0.3990  0.83500  0.416 -0.4220
10997  ZZEF1 -0.1230 -0.0757  0.320  ... -0.0959  0.16900  0.273 -0.0931
10998   ZZZ3 -0.0859 -0.4730 -0.419  ... -0.0635 -0.00809 -0.658  0.0557

[3 rows x 154 columns]

Descriptive Statistics (df.describe(include='all')):
          idx         S001        S002  ...         S151        S152
↪  S153
count  10999  9689.000000  9943.000000  ...  9646.000000  9646.000000
↪  9648.000000
unique  10999         NaN         NaN  ...         NaN         NaN
↪  NaN
top     ZZZ3         NaN         NaN  ...         NaN         NaN
↪  NaN
freq        1         NaN         NaN  ...         NaN         NaN
↪  NaN
mean      NaN   -0.021396   -0.036661  ...    0.079756    0.035783
↪  -0.006398
std       NaN    0.611966    0.678262  ...    0.542243    0.660306
↪  0.528082
min       NaN   -3.550000   -8.190000  ...   -3.220000   -7.050000
↪  -4.190000
25%       NaN   -0.340000   -0.381000  ...   -0.187000   -0.249750
↪  -0.216000
50%       NaN   -0.018600   -0.008360  ...    0.013300    0.007760
↪  0.005690
75%       NaN    0.309000    0.311000  ...    0.272000    0.305000
↪  0.225250
max       NaN    3.570000    5.290000  ...    8.340000    7.490000
↪  2.970000

[11 rows x 154 columns]

Number of Unique Values per Column (df.nunique()):
idx     10999
S001     3088
S002     3207
S003     3146
S004     3062
         ...
S149     3213
S150     3255
S151     3201
S152     3232
S153     3284
Length: 154, dtype: int64

-------------------- Next Sheet --------------------

Analyzing Sheet: 'B-phospho-proteomics' (Sheet 3/3)
-------------------------------------
Shape: 73212 rows, 154 columns
```

```
Column Names:
['idx', 'S001', 'S002', 'S003', 'S004', 'S005', 'S006', 'S007', 'S008',
 ↪  'S009', 'S010', 'S011', 'S012', 'S013', 'S014', 'S015', 'S016',
 ↪  'S017', 'S018', 'S019', 'S020', 'S021', 'S022', 'S023', 'S024',
 ↪  'S025', 'S026', 'S027', 'S028', 'S029', 'S030', 'S031', 'S032',
 ↪  'S033', 'S034', 'S035', 'S036', 'S037', 'S038', 'S039', 'S040',
 ↪  'S041', 'S042', 'S043', 'S044', 'S045', 'S046', 'S047', 'S048',
 ↪  'S049', 'S050', 'S051', 'S052', 'S053', 'S054', 'S055', 'S056',
 ↪  'S057', 'S058', 'S059', 'S060', 'S061', 'S062', 'S063', 'S064',
 ↪  'S065', 'S066', 'S067', 'S068', 'S069', 'S070', 'S071', 'S072',
 ↪  'S073', 'S074', 'S075', 'S076', 'S077', 'S078', 'S079', 'S080',
 ↪  'S081', 'S082', 'S083', 'S084', 'S085', 'S086', 'S087', 'S088',
 ↪  'S089', 'S090', 'S091', 'S092', 'S093', 'S094', 'S095', 'S096',
 ↪  'S097', 'S098', 'S099', 'S100', 'S101', 'S102', 'S103', 'S104',
 ↪  'S105', 'S106', 'S107', 'S108', 'S109', 'S110', 'S111', 'S112',
 ↪  'S113', 'S114', 'S115', 'S116', 'S117', 'S118', 'S119', 'S120',
 ↪  'S121', 'S122', 'S123', 'S124', 'S125', 'S126', 'S127', 'S128',
 ↪  'S129', 'S130', 'S131', 'S132', 'S133', 'S134', 'S135', 'S136',
 ↪  'S137', 'S138', 'S139', 'S140', 'S141', 'S142', 'S143', 'S144',
 ↪  'S145', 'S146', 'S147', 'S148', 'S149', 'S150', 'S151', 'S152',
 ↪  'S153']

Data Types and Non-Null Counts (df.info()):
<class 'pandas.core.frame.DataFrame'>
RangeIndex: 73212 entries, 0 to 73211
Data columns (total 154 columns):
 #    Column  Dtype
---   ------  -----
 0    idx     object
 1    S001    float64
 2    S002    float64
 3    S003    float64
 4    S004    float64
 5    S005    float64
 6    S006    float64
 7    S007    float64
 8    S008    float64
 9    S009    float64
 10   S010    float64
 11   S011    float64
 12   S012    float64
 13   S013    float64
 14   S014    float64
 15   S015    float64
 16   S016    float64
 17   S017    float64
 18   S018    float64
 19   S019    float64
 20   S020    float64
 21   S021    float64
 22   S022    float64
 23   S023    float64
 24   S024    float64
 25   S025    float64
 26   S026    float64
 27   S027    float64
 28   S028    float64
 29   S029    float64
 30   S030    float64
 31   S031    float64
 32   S032    float64
 33   S033    float64
 34   S034    float64
 35   S035    float64
```

```
36    S036    float64
37    S037    float64
38    S038    float64
39    S039    float64
40    S040    float64
41    S041    float64
42    S042    float64
43    S043    float64
44    S044    float64
45    S045    float64
46    S046    float64
47    S047    float64
48    S048    float64
49    S049    float64
50    S050    float64
51    S051    float64
52    S052    float64
53    S053    float64
54    S054    float64
55    S055    float64
56    S056    float64
57    S057    float64
58    S058    float64
59    S059    float64
60    S060    float64
61    S061    float64
62    S062    float64
63    S063    float64
64    S064    float64
65    S065    float64
66    S066    float64
67    S067    float64
68    S068    float64
69    S069    float64
70    S070    float64
71    S071    float64
72    S072    float64
73    S073    float64
74    S074    float64
75    S075    float64
76    S076    float64
77    S077    float64
78    S078    float64
79    S079    float64
80    S080    float64
81    S081    float64
82    S082    float64
83    S083    float64
84    S084    float64
85    S085    float64
86    S086    float64
87    S087    float64
88    S088    float64
89    S089    float64
90    S090    float64
91    S091    float64
92    S092    float64
93    S093    float64
94    S094    float64
95    S095    float64
96    S096    float64
97    S097    float64
98    S098    float64
99    S099    float64
100   S100    float64
```

```
 101  S101     float64
 102  S102     float64
 103  S103     float64
 104  S104     float64
 105  S105     float64
 106  S106     float64
 107  S107     float64
 108  S108     float64
 109  S109     float64
 110  S110     float64
 111  S111     float64
 112  S112     float64
 113  S113     float64
 114  S114     float64
 115  S115     float64
 116  S116     float64
 117  S117     float64
 118  S118     float64
 119  S119     float64
 120  S120     float64
 121  S121     float64
 122  S122     float64
 123  S123     float64
 124  S124     float64
 125  S125     float64
 126  S126     float64
 127  S127     float64
 128  S128     float64
 129  S129     float64
 130  S130     float64
 131  S131     float64
 132  S132     float64
 133  S133     float64
 134  S134     float64
 135  S135     float64
 136  S136     float64
 137  S137     float64
 138  S138     float64
 139  S139     float64
 140  S140     float64
 141  S141     float64
 142  S142     float64
 143  S143     float64
 144  S144     float64
 145  S145     float64
 146  S146     float64
 147  S147     float64
 148  S148     float64
 149  S149     float64
 150  S150     float64
 151  S151     float64
 152  S152     float64
 153  S153     float64
dtypes: float64(153), object(1)
memory usage: 86.0+ MB

First 5 rows (Head):
         idx   S001    S002    S003   S004  ...    S149    S150     S151
  ↪   S152     S153
0  AAAS-S495    NaN     NaN  -0.202  0.25  ...  -0.272  0.223  -0.3940
  ↪   0.149  0.0774
1  AAAS-S541    NaN     NaN     NaN   NaN  ...  -0.191  -0.517  -0.0108
  ↪   0.256  -0.1580
2  AAAS-Y485    NaN     NaN     NaN   NaN  ...     NaN    NaN      NaN
  ↪   NaN      NaN
```

```
3  AACS-S618 -0.881    NaN    NaN   NaN  ...    NaN    NaN    NaN
↪  NaN     NaN
4  AAED1-S12 -1.810  0.084 -1.880   NaN  ...  0.631  0.522  1.0600
↪  0.951 -0.3430

[5 rows x 154 columns]

Last 3 rows (Tail):
             idx   S001    S002  S003  S004  ...  S149    S150    S151
↪  S152    S153
73209  ZZZ3-T415   NaN     NaN   NaN   NaN  ...   NaN    NaN    NaN
↪  NaN     NaN
73210  ZZZ3-T418   NaN  0.1605   NaN   NaN  ...   NaN    NaN    NaN
↪  NaN     NaN
73211  ZZZ3-Y399   NaN -0.0635   NaN   NaN  ...   0.0  0.179 -0.122
↪  -0.354  0.0216

[3 rows x 154 columns]

Descriptive Statistics (df.describe(include='all')):
             idx         S001  ...           S152          S153
count      73212  31671.000000  ...  34285.000000  34303.000000
unique     73212          NaN  ...           NaN           NaN
top    ZZZ3-Y399          NaN  ...           NaN           NaN
freq           1          NaN  ...           NaN           NaN
mean         NaN    -0.065974  ...      0.005505      0.001946
std          NaN     0.641707  ...      0.674115      0.543703
min          NaN    -7.020000  ...     -5.950000     -5.060000
25%          NaN    -0.399000  ...     -0.328000     -0.253000
50%          NaN    -0.030100  ...     -0.001340      0.000000
75%          NaN     0.292000  ...      0.331500      0.261500
max          NaN     4.320000  ...      6.520000      3.940000

[11 rows x 154 columns]

Number of Unique Values per Column (df.nunique()):
idx     73212
S001     7154
S002     7644
S003     5834
S004     6596
         ...
S149     7474
S150     7612
S151     7327
S152     7393
S153     7501
Length: 154, dtype: int64

============================================================
Excel file 'data/mmc2.xlsx' analysis complete.
'''
```

Listing 5: Description of '1-s2.0-S0092867420301070-mmc2.xlsx' in KramaBench.

Moreover, to show that DS-STAR's analyzer agent is capable to generate an accurate and complete text summary even for highly complex and deeply nested data, we first created a dummy xlsx file that contains three tables which are placed in random position. This requires DS-STAR to find the bounding boxes of each table exactly. As shown in the below Listing, we found that DS-STAR successfully handles such complex data, showing the robustness of our analyzer agent.

```
f'''
============================================================
  Excel File Analysis: data/dummy.xlsx
============================================================

Found 1 sheet(s): ['Sheet1']

----------------------------------------------------
  Analyzing Sheet: 'Sheet1'
----------------------------------------------------
- Sheet Dimensions: 15 rows, 11 columns

>>> Found 3 distinct data block(s).

--- Block 1 (Bounding Box: B1:D6) ---
    * Type: Structured Table
    * Shape: 5 rows, 3 columns
    * Column Names: ['exchange', 'ticker', 'name']
    * Data Types:
      - exchange: object
      - ticker: object
      - name: object
    * Data Preview (first 5 rows):
exchange ticker                                          name
  nasdaq   PYPL                               PayPal Holdings
  nasdaq   AMZN                               Amazon.com Inc
  nasdaq   MSFT                               Microsoft Corp.
    nyse    IBM International Business Machines Corporation
  nasdaq  GOOGL                                      Alphabet

--- Block 2 (Bounding Box: I5:K10) ---
    * Type: Structured Table
    * Shape: 5 rows, 3 columns
    * Column Names: ['role', 'age', 'sex']
    * Data Types:
      - role: object
      - age: float64
      - sex: object
    * Data Preview (first 5 rows):
     role  age     sex
passenger 24.0 female
passenger 31.0    male
   driver 72.0 female
   driver 66.0    male
passenger 65.0    male

--- Block 3 (Bounding Box: E10:G15) ---
    * Type: Structured Table
    * Shape: 5 rows, 3 columns
    * Column Names: ['id', 'country_id', 'name']
    * Data Types:
      - id: float64
      - country_id: object
      - name: object
    * Data Preview (first 5 rows):
    id              country_id    name
   1.0 Belgium Jupiler League Belgium
1729.0 England Premier League England
4769.0         France Ligue 1  France
```

```
 7809.0  Germany 1. Bundesliga Germany
10257.0          Italy Serie A   Italy

=========================================================
  End of Analysis
=========================================================
'''
```

Listing 6: Description of 'dummy.xlsx'.

## F QUALITATIVE RESULTS FOR EACH TASK TYPE

This section provides qualitative results. It includes queries on DA-Code benchmark, solution code generated by DS-STAR, output generated by DS-STAR, and ground-truth answer. Specifically, it presents examples for each task, such as statistical analysis, data insights, data wrangling, machine learning, and visualization tasks.

### F.1 STATISTICAL ANALYSIS

```
f'''Could you analyze our Business Case dataset to perform a hypothesis
↪   test comparing bike rentals across different weather conditions? I
↪   need the results formatted and saved in a file named 'weather.csv'
↪   following the requirement in tips.md.'''
```

Listing 7: **User's question.**

```python
import pandas as pd
from scipy.stats import kruskal
import os

# Define the file path for the input dataset
input_file_path = 'data/yulu_bike_sharing_dataset.csv'

# Load the dataset into a pandas DataFrame
yulu_df = pd.read_csv(input_file_path)

# The goal is to perform a Kruskal-Wallis H-test to compare the distribution of bike rentals ('count')
# across different weather conditions ('weather').

# The 'weather' column contains numerical codes for different weather conditions.
# First, we need to find the unique weather conditions in the dataset.
unique_weather_conditions = sorted(yulu_df['weather'].unique())

# Next, we create a list of data samples, where each sample is an array of 'count'
# values corresponding to a specific weather condition.
weather_groups = [yulu_df['count'][yulu_df['weather'] == weather] for weather in unique_weather_conditions]

# Perform the Kruskal-Wallis H-test. The * operator unpacks the list of groups
# into separate arguments for the function.
statistic, p_value = kruskal(*weather_groups)

# Define the significance level (alpha) as specified in the guidelines.
alpha = 0.05

# Based on the p-value, determine the hypothesis conclusion and the comment
# as per the format requirements in 'data/tips.md'.
if p_value < alpha:
    hypothesis_conclusion = "Strong Evidence to Reject Null Hypothesis"
    comment_on_results = "Different distributions"
else:
    hypothesis_conclusion = "No Evidence to Reject Null Hypothesis"
    comment_on_results = "No evidence of different distributions"

# Create a dictionary containing the results in the required format.
results_data = {
    'Test Type': ["Non Parametric"],
    'Kruskal Statistic': [statistic],
    'P-Value': [p_value],
    'Hypothesis': [hypothesis_conclusion],
    'Comment': [comment_on_results]
}

# Create a new pandas DataFrame from the results dictionary.
weather_results_df = pd.DataFrame(results_data)

# Define the output directory and file path.
output_dir = 'final'
output_file_path = os.path.join(output_dir, 'weather.csv')

# Ensure the output directory exists.
os.makedirs(output_dir, exist_ok=True)

# Save the resulting DataFrame to the specified CSV file,
# ensuring the index is not written to the file.
weather_results_df.to_csv(output_file_path, index=False)

# print(f"Analysis complete. Results saved to {output_file_path}")
# print(weather_results_df.to_string())
```

Listing 8: **Solution code.**

Table 7: **Generated output** by DS-STAR in csv format.

| Test Type | Kruskal Statistic | P-Value | Hypothesis | Comment |
|---|---|---|---|---|
| Non Parametric | 205.00216514479087 | 3.501611300708679e-44 | Strong Evidence to Reject Null Hypothesis | Different distributions |

## F.2 DATA INSIGHT

```python
f'''Could you please analyze the matches between Argentina and Brazil?
↪   Record the total number of matches, the number of wins for each
↪   team, and the number of draws. Indicate if Brazil has won more
↪   matches ("Yes" or "No"). Present your findings in the following
↪   format:

"""
{
    "Total Matches between Argentina and Brazil": [...],
    "Arg Wins": [...],
    "Brazil Wins": [...],
    "Draws": [...],
    "Result": [...]
}
"""'''
```

Listing 9: **User's question.**

```python
import pandas as pd
import numpy as np
import json
import os

# Ensure the 'final' directory exists for the output file
output_dir = 'final'
os.makedirs(output_dir, exist_ok=True)

# Load the results.csv file into a pandas DataFrame
results_df = pd.read_csv('data/results.csv')

# Filter for matches where the home team is Argentina and the away team is Brazil
arg_vs_bra = results_df[
    (results_df['home_team'] == 'Argentina') & (results_df['away_team'] == 'Brazil')
]

# Filter for matches where the home team is Brazil and the away team is Argentina
bra_vs_arg = results_df[
    (results_df['home_team'] == 'Brazil') & (results_df['away_team'] == 'Argentina')
]

# Concatenate the two DataFrames to get all matches between Argentina and Brazil
argentina_brazil_matches = pd.concat([arg_vs_bra, bra_vs_arg])

# Sort the combined DataFrame by date for a chronological view
argentina_brazil_matches = argentina_brazil_matches.sort_values(by='date').copy()

# Define the conditions for determining the winner based on scores
# A draw in regular/extra time is considered a draw, regardless of a penalty shootout outcome.
conditions = [
    argentina_brazil_matches['home_score'] > argentina_brazil_matches['away_score'],
    argentina_brazil_matches['away_score'] > argentina_brazil_matches['home_score']
]

# Define the corresponding outcomes (the winning team's name)
choices = [
    argentina_brazil_matches['home_team'],
    argentina_brazil_matches['away_team']
]

# Create the 'winner' column using np.select, with 'Draw' as the default
argentina_brazil_matches['winner'] = np.select(conditions, choices, default='Draw')

# Count the total wins for each team and the number of draws
win_counts = argentina_brazil_matches['winner'].value_counts()

# Calculate the total number of matches, wins for each team, and draws from the win_counts Series
total_matches = int(win_counts.sum())
argentina_wins = int(win_counts.get('Argentina', 0))
brazil_wins = int(win_counts.get('Brazil', 0))
draws = int(win_counts.get('Draw', 0))

# Determine the 'Result' value. 'Yes' if Brazil has more wins, otherwise 'No'.
```

```
result_value = 'Yes' if brazil_wins > argentina_wins else 'No'

# Construct the dictionary to store the analysis results with the specified keys
analysis_results = {
    "Total Matches between Argentina and Brazil": total_matches,
    "Arg Wins": argentina_wins,
    "Brazil Wins": brazil_wins,
    "Draws": draws,
    "Result": result_value
}

# Define the output file path
output_path = os.path.join(output_dir, 'result.json')

# Save the final results dictionary to a JSON file in the 'final' directory
with open(output_path, 'w') as f:
    json.dump(analysis_results, f, indent=4)
```

Listing 10: **Solution code.**

```
f'''{
    "Total Matches between Argentina and Brazil": [109],
    "Arg Wins": [40],
    "Brazil Wins": [43],
    "Draws": [26],
    "Result": ["Yes"]
}'''
```

Listing 11: **Generated output** by DS-STAR, in json format.

### F.3 DATA MANIPULATION

```
f'''Calculate the total quantity sold and total sales revenue for each
↪    bike category from 2016 to 2018. Ensure that total sales revenue
↪    accounts for any discounts applied to each order. Write the results
↪    into result.csv, following the exact structure and formatting of
↪    sample_result.csv.'''
```

Listing 12: **User's question.**

```
import pandas as pd
import os

# Ensure the output directory exists
os.makedirs('final', exist_ok=True)

# Load the required data files into pandas DataFrames
order_items_df = pd.read_csv('data/order_items.csv')
products_df = pd.read_csv('data/products.csv')
categories_df = pd.read_csv('data/categories.csv')
sample_result_df = pd.read_csv('data/sample_result.csv')

# Calculate the sales revenue for each item, accounting for discounts.
# sales_revenue = quantity * list_price * (1 - discount)
order_items_df['sales_revenue'] = order_items_df['quantity'] * order_items_df['list_price'] * (1 -
↪    order_items_df['discount'])

# Merge the order items with product information to get model_year and category_id
merged_df = pd.merge(order_items_df, products_df, on='product_id')

# Merge with category information to get category_name
merged_df = pd.merge(merged_df, categories_df, on='category_id')

# Filter the DataFrame to include only products with model years from 2016 to 2018
filtered_df = merged_df[(merged_df['model_year'] >= 2016) & (merged_df['model_year'] <= 2018)]

# Group the filtered data by 'category_name' and aggregate the total quantity and total sales revenue.
category_summary = filtered_df.groupby('category_name').agg(
    total_quantity=('quantity', 'sum'),
    total_price=('sales_revenue', 'sum')
).reset_index()

# Use the sample_result.csv to define the structure and order of the final output.
# Perform a left merge to ensure all categories from the sample are present in the final result, in the
↪    correct order.
final_df = pd.merge(sample_result_df[['category_name']], category_summary, on='category_name', how='left')

# Fill any potential NaN values with 0 for categories that had no sales in the filtered period.
final_df[['total_quantity', 'total_price']] = final_df[['total_quantity', 'total_price']].fillna(0)
```

```
# Ensure the column order matches the sample_result.csv file: ['total_quantity', 'total_price',
↪  'category_name']
final_df = final_df[['total_quantity', 'total_price', 'category_name']]

# Convert the 'total_quantity' column to integer type as it represents a count of items.
final_df['total_quantity'] = final_df['total_quantity'].astype(int)

# Define the output path
output_path = 'final/result.csv'

# Save the final DataFrame to the specified path, without the index.
final_df.to_csv(output_path, index=False)

print(f"Successfully generated '{output_path}'")
print("Content of the generated file:")
print(final_df)
```

Listing 13: **Solution code.**

Table 8: **Generated output** by DS-STAR in csv format.

| total_quantity | total_price | category_name |
|---|---|---|
| 1755 | 2715079.53 | Mountain Bikes |
| 559 | 1665098.49 | Road Bikes |
| 2063 | 995032.62 | Cruisers Bicycles |
| 315 | 916684.78 | Electric Bikes |
| 394 | 711011.84 | Cyclocross Bicycles |
| 813 | 394020.10 | Comfort Bicycles |
| 1179 | 292189.20 | Children Bicycles |

## F.4 DATA WRANGLING

```
f'''Manage missing data in the NYC_Open_Data_Parking_Violations dataset
↪  by either dropping or replacing it. Drop rows if necessary, and drop
↪  columns only if most entries are empty. Replace missing values with
↪  the mean, mode, or other appropriate methods. Save the cleaned data
↪  to 'cleaned_data.csv'.'''
```

Listing 14: **User's question.**

```
import pandas as pd
import numpy as np
import os

# The user prompt mentions cleaning the 'NYC_Open_Data_Parking_Violations' dataset,
# however, the provided data file is 'data/data.csv', which contains automobile data.
# This script will proceed with cleaning the provided 'data/data.csv' file based on the
# general data cleaning guidelines in the prompt.

# Define file paths
input_file = 'data/data.csv'
output_dir = 'final'
output_file = os.path.join(output_dir, 'cleaned_data.csv')

# Create the output directory if it does not exist
if not os.path.exists(output_dir):
    os.makedirs(output_dir)

# Load the data, interpreting '?' as Not a Number (NaN)
df = pd.read_csv(input_file, na_values='?')

# --- Data Cleaning Steps ---

# Guideline: "Drop rows if necessary"
# The 'price' column is crucial for most analyses. Rows with a missing price are dropped.
df.dropna(subset=['price'], inplace=True)

# Guideline: "Replace missing values with the mean, mode, or other appropriate methods"

# Identify columns for mean and mode imputation
# Numerical columns with missing values to be imputed with the mean
numeric_cols_to_impute_mean = ['normalized-losses', 'bore', 'stroke', 'horsepower', 'peak-rpm']
```

```python
# Categorical columns with missing values to be imputed with the mode
categorical_cols_to_impute_mode = ['num-of-doors']

# Impute numerical columns with the mean
for col in numeric_cols_to_impute_mean:
    # Convert column to numeric type. This is necessary because the '?'
    # values caused pandas to read them as 'object' type.
    df[col] = pd.to_numeric(df[col])
    # Calculate the mean of the column (NaNs are automatically excluded)
    mean_val = df[col].mean()
    # Fill the missing (NaN) values with the calculated mean
    df[col].fillna(mean_val, inplace=True)

# Impute categorical columns with the mode
for col in categorical_cols_to_impute_mode:
    # Calculate the mode (most frequent value) of the column.
    # .mode() returns a Series, so we take the first element.
    mode_val = df[col].mode()[0]
    # Fill the missing (NaN) values with the calculated mode
    df[col].fillna(mode_val, inplace=True)

# Guideline: After handling missing values, ensure data types are correct.
# The 'price' column was loaded as 'object' and needs to be converted to a numeric type.
df['price'] = df['price'].astype(float)

# --- Save the Cleaned Data ---

# Save the final cleaned DataFrame to 'final/cleaned_data.csv'
# The index is not saved as it's not part of the original data.
df.to_csv(output_file, index=False)

# --- Verification ---
# This part confirms that the cleaning was successful and the file was created.
print(f"Successfully cleaned the dataset and saved it to '{output_file}'")

# Load the cleaned data to verify its integrity
cleaned_df = pd.read_csv(output_file)

print("\n--- Verification of Cleaned Data ---")
print(f"Shape of the cleaned data: {cleaned_df.shape}")

# Check for any remaining missing values
missing_values_summary = cleaned_df.isnull().sum()
total_missing = missing_values_summary.sum()

print("\nMissing values per column in the cleaned file:")
print(missing_values_summary)
print(f"\nTotal remaining missing values: {total_missing}")

if total_missing == 0:
    print("\nVerification successful: The cleaned dataset has no missing values.")
else:
    print("\nVerification failed: The cleaned dataset still contains missing values.")
```

Listing 15: **Solution code.**

Table 9: **Generated output** by DS-STAR in csv format.

| symboling | normalized-losses | make | fuel-type | aspiration | num-of-doors | body-style | drive-wheels | engine-location | wheel-base | length | width | height | curb-weight | engine-type | num-of-cylinders | engine-size | fuel-system | bore | stroke | compression-ratio | horsepower | peak-rpm | city-mpg | highway-mpg | price |
|---|---|---|---|---|---|---|---|---|---|---|---|---|---|---|---|---|---|---|---|---|---|---|---|---|---|
| 3 | 122.0 | alfa-romero | gas | std | two | convertible | rwd | front | 88.6 | 168.8 | 64.1 | 48.8 | 2548 | dohc | four | 130 | mpfi | 3.47 | 2.68 | 9.0 | 111 | 5000 | 21 | 27 | 13495 |
| 3 | 122.0 | alfa-romero | gas | std | two | convertible | rwd | front | 88.6 | 168.8 | 64.1 | 48.8 | 2548 | dohc | four | 130 | mpfi | 3.47 | 2.68 | 9.0 | 111 | 5000 | 21 | 27 | 16500 |
| 1 | 122.0 | alfa-romero | gas | std | two | hatchback | rwd | front | 94.5 | 171.2 | 65.5 | 52.4 | 2823 | ohcv | six | 152 | mpfi | 2.68 | 3.47 | 9.0 | 154 | 5000 | 19 | 26 | 16500 |
| ... | ... | ... | ... | ... | ... | ... | ... | ... | ... | ... | ... | ... | ... | ... | ... | ... | ... | ... | ... | ... | ... | ... | ... | ... | ... |

## F.5 MACHINE LEARNING

```python
f'''As a contestant in the Bank customer data churn prediction
    competition, you need to predict the data in test.csv according to
    the competition requirements. Write the results into submission.csv
    following the format specified in sample_submission.csv.'''
```

Listing 16: **User's question.**

```python
import pandas as pd
import numpy as np
from sklearn.model_selection import StratifiedKFold
import lightgbm as lgb
from sklearn.metrics import roc_auc_score
import optuna
import os

# Create the 'final' directory if it doesn't already exist to store the output.
```

```
1998   os.makedirs('final', exist_ok=True)
1999
2000   # Suppress Optuna's logging to keep the output clean during the study.
       optuna.logging.set_verbosity(optuna.logging.WARNING)
2001
2002   # 1. Load All Data
       # Load data from the 'data/' directory.
2003   original_train_df = pd.read_csv('data/train.csv')
       test_df = pd.read_csv('data/test.csv')
2004   additional_data_df = pd.read_csv('data/Churn_Modelling.csv')
2005   # The provided test.csv uses 'CustomerId' as the unique identifier.
       # We will use 'CustomerId' for the submission file's 'id' column as required by the competition format.
2006   test_ids = test_df['CustomerId']
2007
2008   # 2. Preprocess Data: Augment and Clean
       # Select a consistent set of feature columns and the target variable ('Exited') to ensure uniformity across
2009   ↪  datasets.
       feature_cols = ['CreditScore', 'Geography', 'Gender', 'Age', 'Tenure', 'Balance', 'NumOfProducts',
2010   ↪  'HasCrCard', 'IsActiveMember', 'EstimatedSalary', 'Exited']
2011
       # Augment the training data by combining the original training set with the additional 'Churn_Modelling.csv'
2012   ↪  dataset.
       train_df = pd.concat([original_train_df[feature_cols], additional_data_df[feature_cols]], ignore_index=True)
2013
2014
       # --- START: Feature Engineering ---
2015   def feature_engineer(df):
           """Creates new features based on domain knowledge and interactions between existing features."""
2016       # A small epsilon is added to denominators to prevent division by zero errors.
           epsilon = 1e-6
2017       df['BalanceSalaryRatio'] = df['Balance'] / (df['EstimatedSalary'] + epsilon)
           df['TenureByAge'] = df['Tenure'] / (df['Age'] + epsilon)
2018       df['CreditScoreGivenAge'] = df['CreditScore'] / (df['Age'] + epsilon)
           df['Loyalty'] = df['Tenure'] * df['IsActiveMember']
2019       df['ProductsPerTenure'] = df['NumOfProducts'] / (df['Tenure'] + epsilon)
           return df
2020
2021   print("Applying feature engineering to training and test data...")
2022   train_df = feature_engineer(train_df)
       test_df = feature_engineer(test_df)
2023   # --- END: Feature Engineering ---
2024
       # Drop identifier columns ('CustomerId', 'Surname', 'id') that are not useful for the model.
2025   # 'errors='ignore'' handles cases where a column might not exist in a dataframe.
       drop_cols = ['CustomerId', 'Surname', 'id']
2026   train_df = train_df.drop(columns=[col for col in drop_cols if col in train_df.columns and col != 'Exited'],
2027   ↪  errors='ignore')
       test_df = test_df.drop(columns=[col for col in drop_cols if col in test_df.columns], errors='ignore')
2028
2029
       # One-Hot Encode categorical features ('Geography', 'Gender') to convert them into a machine-readable
2030   ↪  format.
       categorical_features = ['Geography', 'Gender']
2031   train_df = pd.get_dummies(train_df, columns=categorical_features, drop_first=True)
2032   test_df = pd.get_dummies(test_df, columns=categorical_features, drop_first=True)
2033   # Separate the target variable ('Exited') from the training features.
       train_labels = train_df['Exited']
2034   train_df = train_df.drop(columns=['Exited'])
2035   # Align the training and test dataframes to ensure they have the exact same feature columns in the same
2036   ↪  order.
       # 'inner' join keeps only columns that are present in both dataframes.
2037   train_df, test_df = train_df.align(test_df, join='inner', axis=1)
2038   # Define the final feature set (X) and target (y).
2039   X = train_df
       y = train_labels
2040
       # 3. Define the Objective Function for Optuna Hyperparameter Tuning
2041   def objective(trial):
           """Defines the hyperparameter search space and evaluation metric for Optuna to optimize."""
2042       # Define the hyperparameter search space for the LightGBM model.
       params = {
2043           'objective': 'binary',
2044           'metric': 'auc',
               'boosting_type': 'gbdt',
2045           'n_estimators': trial.suggest_int('n_estimators', 2000, 10000),
2046           'learning_rate': trial.suggest_float('learning_rate', 0.01, 0.05),
               'num_leaves': trial.suggest_int('num_leaves', 20, 100),
2047           'max_depth': trial.suggest_int('max_depth', 5, 10),
               'seed': 42,
2048           'n_jobs': -1,
               'verbose': -1,
2049           'colsample_bytree': trial.suggest_float('colsample_bytree', 0.5, 0.9),
2050           'subsample': trial.suggest_float('subsample', 0.5, 0.9),
               'reg_alpha': trial.suggest_float('reg_alpha', 0.01, 0.5),
2051           'reg_lambda': trial.suggest_float('reg_lambda', 0.01, 0.5),
       }
```

```
N_SPLITS = 5
skf = StratifiedKFold(n_splits=N_SPLITS, shuffle=True, random_state=42)
oof_auc_scores = []

# Perform cross-validation to get a robust estimate of the model's performance for a given set of
↪  hyperparameters.
for fold, (train_idx, val_idx) in enumerate(skf.split(X, y)):
    X_train, y_train = X.iloc[train_idx], y.iloc[train_idx]
    X_val, y_val = X.iloc[val_idx], y.iloc[val_idx]

    model = lgb.LGBMClassifier(**params)
    model.fit(X_train, y_train,
              eval_set=[(X_val, y_val)],
              eval_metric='auc',
              callbacks=[lgb.early_stopping(100, verbose=False)])

    val_preds = model.predict_proba(X_val)[:, 1]
    auc = roc_auc_score(y_val, val_preds)
    oof_auc_scores.append(auc)

# Return the mean AUC score, which Optuna will try to maximize.
return np.mean(oof_auc_scores)

# 4. Run the Optimization Study
print("Running hyperparameter optimization...")
study = optuna.create_study(direction='maximize')
# The number of trials can be adjusted; 50 provides a reasonable balance between search time and
↪  performance.
study.optimize(objective, n_trials=50)
best_params = study.best_params
print(f"\nBest trial parameters: {best_params}")
print(f"Best trial mean OOF AUC: {study.best_value}")

# 5. Train Final Model with Best Parameters and Create Submission
print("\nTraining final model with tuned parameters...")
N_SPLITS = 5
skf = StratifiedKFold(n_splits=N_SPLITS, shuffle=True, random_state=42)
test_preds = np.zeros(len(test_df))
oof_auc_scores = []

# Combine the best parameters found by Optuna with fixed model settings.
final_params = {
    'objective': 'binary', 'metric': 'auc', 'boosting_type': 'gbdt',
    'seed': 42, 'n_jobs': -1, 'verbose': -1, **best_params
}

for fold, (train_idx, val_idx) in enumerate(skf.split(X, y)):
    print(f"===== Fold {fold+1} =====")
    X_train, y_train = X.iloc[train_idx], y.iloc[train_idx]
    X_val, y_val = X.iloc[val_idx], y.iloc[val_idx]

    model = lgb.LGBMClassifier(**final_params)
    model.fit(X_train, y_train,
              eval_set=[(X_val, y_val)],
              eval_metric='auc',
              callbacks=[lgb.early_stopping(100, verbose=False)])

    val_preds = model.predict_proba(X_val)[:, 1]
    auc = roc_auc_score(y_val, val_preds)
    oof_auc_scores.append(auc)
    print(f"Fold {fold+1} AUC: {auc}")

    # Accumulate predictions on the test set, averaging over the folds to improve robustness.
    test_preds += model.predict_proba(test_df)[:, 1] / N_SPLITS

# Print the final average validation score from the cross-validation.
print(f"\nAverage OOF AUC with tuned parameters: {np.mean(oof_auc_scores)}")

# Create the submission DataFrame using the averaged test predictions.
submission_df = pd.DataFrame({'id': test_ids, 'Exited': test_preds})

# Save the DataFrame to a CSV file in the 'final' directory for submission.
submission_df.to_csv('final/submission.csv', index=False)

print("\nSubmission file created successfully at 'final/submission.csv'.")
print("First 5 rows of the submission file:")
print(submission_df.head())
```

Listing 17: **Solution code and machine learning model.**

## F.6 VISUALIZATION

```
f'''Draw a stacked bar chart that displays the percentage of restaurants
↪    offering online ordering options versus those not offering online
↪    ordering options across different rating levels. Save this chart as
↪    'result.jpg'. The title of the image should be "Percentage of
↪    Restaurants' Online Order Option by Rating", with the xlabel as
↪    "Rating", and the ylabel as "Percentage of Online Orders".'''
```

Listing 18: **User's question.**

```python
import pandas as pd
import sqlite3
import zipfile
import os
import numpy as np
import matplotlib.pyplot as plt

def generate_chart():
    """
    Loads data from the Zomato SQLite database, cleans the 'rate' column,
    calculates the percentage of restaurants offering online ordering for each rating,
    and generates a stacked bar chart of the results, saving it to the 'final' directory.
    """
    # Define file paths
    zip_path = 'data/zomato_rawdata.sqlite.zip'
    db_name = 'zomato_rawdata.sqlite'
    extract_path = '.'  # Extract to the current directory
    output_dir = 'final'
    output_path = os.path.join(output_dir, 'result.jpg')

    # --- Step 1: Create output directory if it doesn't exist ---
    if not os.path.exists(output_dir):
        os.makedirs(output_dir)

    # --- Step 2: Extract the database from the zip file ---
    db_path = os.path.join(extract_path, db_name)
    with zipfile.ZipFile(zip_path, 'r') as zip_ref:
        # Extract the database file, ensuring the correct member is chosen
        for member in zip_ref.infolist():
            if db_name in member.filename:
                # To handle cases where the zip file has directory structures
                member.filename = os.path.basename(member.filename)
                zip_ref.extract(member, path=extract_path)
                break

    # --- Step 3: Load data from the SQLite database ---
    conn = None
    try:
        conn = sqlite3.connect(db_path)
        # Select only the columns needed for this analysis
        query = "SELECT rate, online_order FROM Users"
        df = pd.read_sql_query(query, conn)
    finally:
        if conn:
            conn.close()

    # --- Step 4: Clean the 'rate' column ---
    # Make a copy to avoid SettingWithCopyWarning
    df = df.copy()

    # Replace special non-numeric strings 'NEW' and '-' with numpy.nan
    df['rate'] = df['rate'].replace(['NEW', '-'], np.nan)

    # Remove the '/5' suffix from the rating strings
    df['rate'] = df['rate'].str.replace('/5', '').str.strip()

    # Convert the cleaned column to a numeric (float) type
    df['rate'] = pd.to_numeric(df['rate'], errors='coerce')

    # Drop rows where 'rate' or 'online_order' is NaN as they are not useful for this analysis
    df.dropna(subset=['rate', 'online_order'], inplace=True)

    # --- Step 5: Aggregate the data for plotting ---
    # Create a cross-tabulation to get counts of 'Yes' and 'No' for each rating
    online_order_counts = pd.crosstab(df['rate'], df['online_order'])

    # Calculate the percentage by dividing each row by its sum
    rate_online_percentage_df = online_order_counts.div(online_order_counts.sum(axis=1), axis=0) * 100

    # --- Step 6: Generate and save the stacked bar chart ---
    # Create the plot figure and axes
    fig, ax = plt.subplots(figsize=(14, 8))

    # Generate the stacked bar chart
    rate_online_percentage_df.plot(kind='bar', stacked=True, ax=ax)
```

```
      # Set the title and labels as required
      ax.set_title("Percentage of Restaurants' Online Order Option by Rating", fontsize=16)
      ax.set_xlabel("Rating", fontsize=12)
      ax.set_ylabel("Percentage of Online Orders", fontsize=12)

      # Improve aesthetics and readability
      ax.tick_params(axis='x', rotation=90)
      ax.legend(title='Online Order')
      ax.grid(axis='y', linestyle='--', alpha=0.7)

      # Adjust layout to prevent labels from being cut off
      plt.tight_layout()

      # Save the resulting chart to the final directory
      plt.savefig(output_path)

      # Close the plot to free up memory
      plt.close(fig)

      print(f"Chart has been generated and saved as '{output_path}'.")

      # --- Step 7: Clean up the extracted file ---
      if os.path.exists(db_path):
          os.remove(db_path)

# Execute the function to generate the chart
generate_chart()
```

Listing 19: **Solution code.**

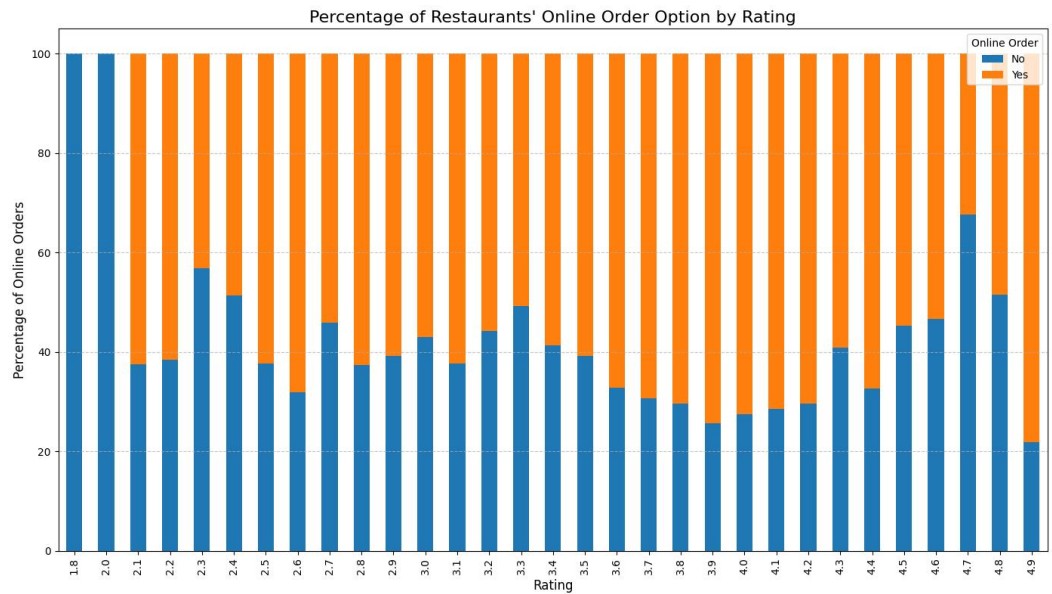

Figure 5: **Generated bar chart.**

# G  FULL EXECUTION LOGS OF DS-STAR

In this section, we provide full logs of DS-STAR to better understand its behavior.

```
f'''In February 2023 what delta would Rafa_AI pay if the relative fee of
↪  the fee with ID=17 changed to 99?'''
```

Listing 20: **User's question.**

```
# Step 1
f'''First, I will filter the `payments.csv` data to select only the
↪  transactions for the merchant 'Rafa_AI' that occurred in February
↪  2023. Since 2023 is not a leap year, February corresponds to the
↪  days of the year from 32 to 59.'''
```

Listing 21: **Round 0: Initial plan.**

```python
import pandas as pd

# Load the payments data
try:
    payments_df = pd.read_csv('data/payments.csv')
except FileNotFoundError:
    print("Error: 'data/payments.csv' not found. Please ensure the file is in the correct directory.")
    exit()

# Define the filter for the merchant 'Rafa_AI'
merchant_filter = payments_df['merchant'] == 'Rafa_AI'

# Define the filter for February 2023.
# 2023 is not a leap year. January has 31 days.
# February runs from day 32 (31+1) to day 59 (31+28).
date_filter = (payments_df['day_of_year'] >= 32) & (payments_df['day_of_year'] <= 59)

# Apply both filters to the DataFrame
rafa_ai_feb_transactions = payments_df[merchant_filter & date_filter]

# Display the resulting filtered DataFrame
print(rafa_ai_feb_transactions)
```

Listing 22: **Round 0: Initial implementation.**

```
f'''Insufficient'''
```

Listing 23: **Round 0: Validation.**

```
f'''Add Step'''
```

Listing 24: **Round 0: Routing.** Add Step 2 to the initial plan.

```
# Step 1
f'''First, I will filter the `payments.csv` data to select only the
↪  transactions for the merchant 'Rafa_AI' that occurred in February
↪  2023. Since 2023 is not a leap year, February corresponds to the
↪  days of the year from 32 to 59.'''

# Step 2
f'''I will calculate the original total fee paid by Rafa_AI in February
↪  2023. This involves loading the `merchant_data.json` and `fees.json`
↪  files, enriching the transaction data with all attributes required
↪  for fee matching (including merchant details, monthly volume,
↪  monthly fraud rate, and intracountry status), then applying the
↪  corresponding fee rule to each transaction and summing the
↪  calculated fees..'''
```

Listing 25: **Round 1: Plan.**

```python
import pandas as pd
import json

# --- 1. Load All Necessary Data ---
try:
    payments_df = pd.read_csv('data/payments.csv')
    merchant_data_df = pd.read_json('data/merchant_data.json')
    fees_df = pd.read_json('data/fees.json')
    acquirer_countries_df = pd.read_csv('data/acquirer_countries.csv')
except FileNotFoundError as e:
    print(f"Error loading data: {e}. Please ensure all data files are in the 'data/' directory.")
    exit()

# --- 2. Filter Transactions (Based on Previous Plan) ---
# Define the filter for the merchant 'Rafa_AI'
merchant_filter = payments_df['merchant'] == 'Rafa_AI'

# Define the filter for February 2023.
# 2023 is not a leap year. January has 31 days.
# February runs from day 32 (31+1) to day 59 (31+28).
date_filter = (payments_df['day_of_year'] >= 32) & (payments_df['day_of_year'] <= 59)

# Apply both filters to the DataFrame and create a copy to avoid warnings
rafa_ai_feb_transactions = payments_df[merchant_filter & date_filter].copy()

# --- 3. Enrich Transaction Data with Merchant and Acquirer Information ---

# Get Rafa_AI's specific merchant data
rafa_ai_merchant_info = merchant_data_df[merchant_data_df['merchant'] == 'Rafa_AI'].iloc[0]

# Add merchant-specific attributes to the transaction DataFrame
rafa_ai_feb_transactions['account_type'] = rafa_ai_merchant_info['account_type']
rafa_ai_feb_transactions['capture_delay'] = str(rafa_ai_merchant_info['capture_delay']) # Ensure consistent
↪  type
rafa_ai_feb_transactions['merchant_category_code'] = rafa_ai_merchant_info['merchant_category_code']

# FIX: The original code failed on the next line.
# The error "ValueError: Length of values (1) does not match length of index (2126)"
# suggests that rafa_ai_merchant_info['acquirer'] is a list-like object with one element
# (e.g., ['gringotts']) instead of a scalar string. Assigning a list of length 1 to a
# column of a different length causes this error.
# The fix is to extract the scalar string from the list before assigning it.
acquirer_name = rafa_ai_merchant_info['acquirer']
if isinstance(acquirer_name, list):
    # If it's a list, take the first element. If empty, it becomes None.
    acquirer_name = acquirer_name[0] if acquirer_name else None

# Now assign the scalar acquirer name to the new column.
rafa_ai_feb_transactions['acquirer'] = acquirer_name

# Create a mapping from acquirer name to country code
acquirer_country_map = acquirer_countries_df.set_index('acquirer')['country_code']

# Add acquirer country to the transaction data
rafa_ai_feb_transactions['acquirer_country'] =
↪  rafa_ai_feb_transactions['acquirer'].map(acquirer_country_map)

# Determine if each transaction is intracountry
rafa_ai_feb_transactions['intracountry'] = rafa_ai_feb_transactions['issuing_country'] ==
↪  rafa_ai_feb_transactions['acquirer_country']

# --- 4. Calculate Monthly Aggregates for Fee Matching ---

# Calculate total transaction volume for Rafa_AI in February
```

```
2322
2323    monthly_volume = rafa_ai_feb_transactions['eur_amount'].sum()

2324    # Calculate fraudulent transaction volume for the same period
        fraudulent_volume =
2325    ↪    rafa_ai_feb_transactions[rafa_ai_feb_transactions['has_fraudulent_dispute']]['eur_amount'].sum()

2326    # Calculate the monthly fraud rate as a percentage
        # Avoid division by zero if there's no volume
2327    monthly_fraud_rate = (fraudulent_volume / monthly_volume) if monthly_volume > 0 else 0
        monthly_fraud_percentage = monthly_fraud_rate * 100
2328
        # Add these aggregates to each transaction row for easy lookup during fee matching
2329    rafa_ai_feb_transactions['monthly_volume'] = monthly_volume
        rafa_ai_feb_transactions['monthly_fraud_percentage'] = monthly_fraud_percentage
2330
2331
2332    # --- 5. Implement Fee Matching Logic ---

2333    def parse_numeric_value(s):
            """Helper to parse strings like '100k' or '1m' into numbers."""
2334        s = s.strip().lower()
            if 'k' in s:
2335            return float(s.replace('k', '')) * 1_000
            if 'm' in s:
2336            return float(s.replace('m', '')) * 1_000_000
            return float(s)
2337
2338    def check_volume_rule(rule_str, value):
2339        """Checks if a value fits a volume rule string (e.g., '100k-1m')."""
            if pd.isnull(rule_str): return True
2340        rule_str = rule_str.replace(' ', '')
            if '-' in rule_str:
2341            low_str, high_str = rule_str.split('-')
                return parse_numeric_value(low_str) <= value < parse_numeric_value(high_str)
2342        if '>' in rule_str:
                return value >= parse_numeric_value(rule_str.replace('>', ''))
2343        if '<' in rule_str:
                return value < parse_numeric_value(rule_str.replace('<', ''))
2344        return False
2345
2346    def check_fraud_rule(rule_str, value):
            """Checks if a value fits a fraud rule string (e.g., '7.7%-8.3%')."""
2347        if pd.isnull(rule_str): return True
            rule_str = rule_str.replace('%', '')
2348        if '-' in rule_str:
                low_str, high_str = rule_str.split('-')
2349            return float(low_str) <= value < float(high_str)
            if '>' in rule_str:
2350            return value >= float(rule_str.replace('>', ''))
            if '<' in rule_str:
2351            return value < float(rule_str.replace('<', ''))
2352        return False
2353
2354    def find_matching_fee_rule(transaction, all_rules_df):
            """Iterates through fee rules to find the first one that matches the transaction."""
2355        for _, rule in all_rules_df.iterrows():
                # A rule with an empty list for a category applies to all values in that category
2356            if rule['card_scheme'] != transaction['card_scheme']: continue
                if rule['account_type'] and transaction['account_type'] not in rule['account_type']: continue
2357            if rule['merchant_category_code'] and transaction['merchant_category_code'] not in
                ↪    rule['merchant_category_code']: continue
2358            if rule['aci'] and transaction['aci'] not in rule['aci']: continue

2359            # A rule with a null value for a category applies to all values
2360            if pd.notnull(rule['is_credit']) and rule['is_credit'] != transaction['is_credit']: continue
                if pd.notnull(rule['intracountry']) and rule['intracountry'] != transaction['intracountry']:
2361            ↪    continue
                if pd.notnull(rule['capture_delay']) and rule['capture_delay'] != transaction['capture_delay']:
2362            ↪    continue
2363
                # Check complex range-based rules
2364            if not check_volume_rule(rule['monthly_volume'], transaction['monthly_volume']): continue
                if not check_fraud_rule(rule['monthly_fraud_level'], transaction['monthly_fraud_percentage']):
2365            ↪    continue
2366
                # If all checks pass, this is the correct rule
2367            return rule['fixed_amount'], rule['rate']

2368        # Return default/zero fees if no rule is found (should not happen with complete data)
        return 0, 0
2369
2370    # --- 6. Calculate Fees and Final Sum ---

2371    # Apply the matching function to each transaction to get its specific fee parameters
        fee_params = rafa_ai_feb_transactions.apply(
2372        lambda row: find_matching_fee_rule(row, fees_df),
            axis=1
2373    )
2374
        # Assign the found fee parameters back to the DataFrame
2375    rafa_ai_feb_transactions[['fixed_amount', 'rate']] = pd.DataFrame(
            fee_params.tolist(),
```

```
     index=rafa_ai_feb_transactions.index
)

# Calculate the fee for each transaction using the formula from the manual
# fee = fixed_amount + rate * transaction_value / 10000
rafa_ai_feb_transactions['fee'] = rafa_ai_feb_transactions['fixed_amount'] + \
                          (rafa_ai_feb_transactions['rate'] * rafa_ai_feb_transactions['eur_amount'] /
                          ↪  10000)

# Calculate the total fee by summing up the fees for all transactions
total_original_fee = rafa_ai_feb_transactions['fee'].sum()

# Display the final result
print(f"The original total fee paid by Rafa_AI in February 2023 was: {total_original_fee:.2f} EUR")
```

Listing 26: **Round 1: Implementation.**

```
f'''Insufficient'''
```

Listing 27: **Round 1: Validation.**

```
f'''Add Step'''
```

Listing 28: **Round 1: Routing.** Add Step 3 to the current plan.

```
# Step 1
f'''First, I will filter the `payments.csv` data to select only the
↪  transactions for the merchant 'Rafa_AI' that occurred in February
↪  2023. Since 2023 is not a leap year, February corresponds to the
↪  days of the year from 32 to 59.'''

# Step 2
f'''I will calculate the original total fee paid by Rafa_AI in February
↪  2023. This involves loading the `merchant_data.json` and `fees.json`
↪  files, enriching the transaction data with all attributes required
↪  for fee matching (including merchant details, monthly volume,
↪  monthly fraud rate, and intracountry status), then applying the
↪  corresponding fee rule to each transaction and summing the
↪  calculated fees..'''

# Step 3
f'''I will calculate the new total fee for Rafa_AI's February 2023
↪  transactions using the modified fee structure where the fee rule
↪  with ID=17 has its `rate` changed to 99. Then, I will subtract the
↪  original total fee (778.52 EUR) from this new total fee to find the
↪  requested delta.'''
```

Listing 29: **Round 2: Plan.**

```
import pandas as pd
import json

# --- 1. Load All Necessary Data ---
try:
    payments_df = pd.read_csv('data/payments.csv')
    merchant_data_df = pd.read_json('data/merchant_data.json')
    fees_df = pd.read_json('data/fees.json')
    acquirer_countries_df = pd.read_csv('data/acquirer_countries.csv')
except FileNotFoundError as e:
    print(f"Error loading data: {e}. Please ensure all data files are in the 'data/' directory.")
    exit()

# --- 2. Filter Transactions (Based on Previous Plan) ---
# Define the filter for the merchant 'Rafa_AI'
merchant_filter = payments_df['merchant'] == 'Rafa_AI'

# Define the filter for February 2023.
# 2023 is not a leap year. January has 31 days.
# February runs from day 32 (31+1) to day 59 (31+28).
```

```
2430
2431        date_filter = (payments_df['day_of_year'] >= 32) & (payments_df['day_of_year'] <= 59)

2432        # Apply both filters to the DataFrame and create a copy to avoid warnings
            rafa_ai_feb_transactions = payments_df[merchant_filter & date_filter].copy()
2433
2434        # --- 3. Enrich Transaction Data with Merchant and Acquirer Information ---

2435        # Get Rafa_AI's specific merchant data
            rafa_ai_merchant_info = merchant_data_df[merchant_data_df['merchant'] == 'Rafa_AI'].iloc[0]
2436
2437        # Add merchant-specific attributes to the transaction DataFrame
            rafa_ai_feb_transactions['account_type'] = rafa_ai_merchant_info['account_type']
2438        rafa_ai_feb_transactions['capture_delay'] = str(rafa_ai_merchant_info['capture_delay']) # Ensure consistent
        ↪   type
2439        rafa_ai_feb_transactions['merchant_category_code'] = rafa_ai_merchant_info['merchant_category_code']

2440        # FIX: The original code failed on the next line.
            # The error "ValueError: Length of values (1) does not match length of index (2126)"
2441        # suggests that rafa_ai_merchant_info['acquirer'] is a list-like object with one element
            # (e.g., ['gringotts']) instead of a scalar string. Assigning a list of length 1 to a
2442        # column of a different length causes this error.
            # The fix is to extract the scalar string from the list before assigning it.
2443        acquirer_name = rafa_ai_merchant_info['acquirer']
            if isinstance(acquirer_name, list):
2444            # If it's a list, take the first element. If empty, it becomes None.
                acquirer_name = acquirer_name[0] if acquirer_name else None
2445
2446        # Now assign the scalar acquirer name to the new column.
2447        rafa_ai_feb_transactions['acquirer'] = acquirer_name

2448
            # Create a mapping from acquirer name to country code
2449        acquirer_country_map = acquirer_countries_df.set_index('acquirer')['country_code']

2450        # Add acquirer country to the transaction data
2451        rafa_ai_feb_transactions['acquirer_country'] =
        ↪   rafa_ai_feb_transactions['acquirer'].map(acquirer_country_map)
2452
            # Determine if each transaction is intracountry
2453        rafa_ai_feb_transactions['intracountry'] = rafa_ai_feb_transactions['issuing_country'] ==
2454    ↪   rafa_ai_feb_transactions['acquirer_country']

2455        # --- 4. Calculate Monthly Aggregates for Fee Matching ---

2456        # Calculate total transaction volume for Rafa_AI in February
2457        monthly_volume = rafa_ai_feb_transactions['eur_amount'].sum()

2458        # Calculate fraudulent transaction volume for the same period
            fraudulent_volume =
2459    ↪   rafa_ai_feb_transactions[rafa_ai_feb_transactions['has_fraudulent_dispute']]['eur_amount'].sum()

2460        # Calculate the monthly fraud rate as a percentage
2461        # Avoid division by zero if there's no volume
            monthly_fraud_rate = (fraudulent_volume / monthly_volume) if monthly_volume > 0 else 0
2462        monthly_fraud_percentage = monthly_fraud_rate * 100

2463        # Add these aggregates to each transaction row for easy lookup during fee matching
2464        rafa_ai_feb_transactions['monthly_volume'] = monthly_volume
            rafa_ai_feb_transactions['monthly_fraud_percentage'] = monthly_fraud_percentage
2465
2466        # --- 5. Implement Fee Matching Logic ---

2467        def parse_numeric_value(s):
2468            """Helper to parse strings like '100k' or '1m' into numbers."""
                s = s.strip().lower()
2469            if 'k' in s:
                    return float(s.replace('k', '')) * 1_000
2470            if 'm' in s:
2471                return float(s.replace('m', '')) * 1_000_000
                return float(s)
2472
            def check_volume_rule(rule_str, value):
2473            """Checks if a value fits a volume rule string (e.g., '100k-1m')."""
                if pd.isnull(rule_str): return True
2474            rule_str = rule_str.replace(' ', '')
                if '-' in rule_str:
2475                low_str, high_str = rule_str.split('-')
                    return parse_numeric_value(low_str) <= value < parse_numeric_value(high_str)
2476            if '>' in rule_str:
                    return value >= parse_numeric_value(rule_str.replace('>', ''))
2477            if '<' in rule_str:
                    return value < parse_numeric_value(rule_str.replace('<', ''))
2478            return False

2479        def check_fraud_rule(rule_str, value):
2480            """Checks if a value fits a fraud rule string (e.g., '7.7%-8.3%')."""
                if pd.isnull(rule_str): return True
2481            rule_str = rule_str.replace('%', '')
                if '-' in rule_str:
2482                low_str, high_str = rule_str.split('-')
                    return float(low_str) <= value < float(high_str)
2483
```

```
2484        if '>' in rule_str:
2485            return value >= float(rule_str.replace('>', ''))
2486        if '<' in rule_str:
2487            return value < float(rule_str.replace('<', ''))
2488        return False

     def find_matching_fee_rule(transaction, all_rules_df):
2489        """Iterates through fee rules to find the first one that matches the transaction."""
2490        for _, rule in all_rules_df.iterrows():
2491            # A rule with an empty list for a category applies to all values in that category
2492            if rule['card_scheme'] != transaction['card_scheme']: continue
2493            if rule['account_type'] and transaction['account_type'] not in rule['account_type']: continue
             if rule['merchant_category_code'] and transaction['merchant_category_code'] not in
             ↪    rule['merchant_category_code']: continue
             if rule['aci'] and transaction['aci'] not in rule['aci']: continue
2494
2495            # A rule with a null value for a category applies to all values
             if pd.notnull(rule['is_credit']) and rule['is_credit'] != transaction['is_credit']: continue
             if pd.notnull(rule['intracountry']) and rule['intracountry'] != transaction['intracountry']:
2496            ↪    continue
             if pd.notnull(rule['capture_delay']) and rule['capture_delay'] != transaction['capture_delay']:
2497            ↪    continue

2498            # Check complex range-based rules
             if not check_volume_rule(rule['monthly_volume'], transaction['monthly_volume']): continue
2499            if not check_fraud_rule(rule['monthly_fraud_level'], transaction['monthly_fraud_percentage']):
2500            ↪    continue

2501            # If all checks pass, this is the correct rule
             return rule['fixed_amount'], rule['rate']

2503        # Return default/zero fees if no rule is found (should not happen with complete data)
         return 0, 0

2504 # --- 6. Calculate Fees and Final Sum ---

2505 # Apply the matching function to each transaction to get its specific fee parameters
2506 fee_params = rafa_ai_feb_transactions.apply(
2507     lambda row: find_matching_fee_rule(row, fees_df),
         axis=1
2508 )

2509 # Assign the found fee parameters back to the DataFrame
2510 rafa_ai_feb_transactions[['fixed_amount', 'rate']] = pd.DataFrame(
         fee_params.tolist(),
2511     index=rafa_ai_feb_transactions.index
2512 )

2513 # Calculate the fee for each transaction using the formula from the manual
     # fee = fixed_amount + rate * transaction_value / 10000
2514 rafa_ai_feb_transactions['fee'] = rafa_ai_feb_transactions['fixed_amount'] + \
                                        (rafa_ai_feb_transactions['rate'] * rafa_ai_feb_transactions['eur_amount'] /
2515                                     ↪    10000)

2516 # Calculate the total fee by summing up the fees for all transactions
2517 total_original_fee = rafa_ai_feb_transactions['fee'].sum()

2518 # Display the final result from the previous step
2519 print(f"The original total fee paid by Rafa_AI in February 2023 was: {total_original_fee:.2f} EUR")

2520 # --- 7. Calculate New Fee with Modified Rule ---

2521 # Create a copy of the fees dataframe to avoid altering the original
2522 fees_df_modified = fees_df.copy()

2523 # Implement the plan: change the rate for the rule with ID=17 to 99
2524 fees_df_modified.loc[fees_df_modified['ID'] == 17, 'rate'] = 99

2525 # Apply the matching function again, this time with the modified fee rules
2526 new_fee_params = rafa_ai_feb_transactions.apply(
         lambda row: find_matching_fee_rule(row, fees_df_modified),
2527     axis=1
     )

2528 # Assign the new fee parameters to new columns in the DataFrame
2529 rafa_ai_feb_transactions[['new_fixed_amount', 'new_rate']] = pd.DataFrame(
         new_fee_params.tolist(),
2530     index=rafa_ai_feb_transactions.index
2531 )

2532 # Calculate the new fee for each transaction using the new rate
     rafa_ai_feb_transactions['new_fee'] = rafa_ai_feb_transactions['new_fixed_amount'] + \
2533                                        (rafa_ai_feb_transactions['new_rate'] *
2534                                     ↪    rafa_ai_feb_transactions['eur_amount'] / 10000)

2535 # Calculate the new total fee by summing up the individual new fees
     new_total_fee = rafa_ai_feb_transactions['new_fee'].sum()
2536
2537 # Calculate the difference (delta) between the new total fee and the original total fee
     fee_delta = new_total_fee - total_original_fee
```

```
# --- 8. Display the Final Delta ---
print(f"The new total fee with the modified rule would be: {new_total_fee:.2f} EUR")
print(f"The requested delta (New Fee - Original Fee) is: {fee_delta:.2f} EUR")
```

Listing 30: **Round 2: Implementation.**

```
f'''Insufficient'''
```

Listing 31: **Round 2: Validation.**

```
f'''Step 3 is wrong'''
```

Listing 32: **Round 2: Routing.** Remove Step 3 from the current plan and add Step 3 again.

```
# Step 1
f'''First, I will filter the `payments.csv` data to select only the
↪  transactions for the merchant 'Rafa_AI' that occurred in February
↪  2023. Since 2023 is not a leap year, February corresponds to the
↪  days of the year from 32 to 59.'''

# Step 2
f'''I will calculate the original total fee paid by Rafa_AI in February
↪  2023. This involves loading the `merchant_data.json` and `fees.json`
↪  files, enriching the transaction data with all attributes required
↪  for fee matching (including merchant details, monthly volume,
↪  monthly fraud rate, and intracountry status), then applying the
↪  corresponding fee rule to each transaction and summing the
↪  calculated fees..'''

# Step 3
f'''I will filter the transactions for 'Rafa_AI' in February 2023
↪  (day_of_year from 32 to 59) that meet all the conditions of fee rule
↪  ID=17: `card_scheme` must be 'SwiftCharge', `is_credit` must be
↪  True, and `aci` must be 'A'. Then, for these matching transactions,
↪  I will calculate the total fee difference using the formula: (99 -
↪  60) * SUM(eur_amount) / 10000.'''
```

Listing 33: **Round 3: Plan.**

```
import pandas as pd
import json

# --- 1. Load All Necessary Data ---
# This part is based on the loading logic from the base code.
try:
    payments_df = pd.read_csv('data/payments.csv')
    merchant_data_df = pd.read_json('data/merchant_data.json')
    fees_df = pd.read_json('data/fees.json')
except FileNotFoundError as e:
    print(f"Error loading data: {e}. Please ensure all data files are in the 'data/' directory.")
    exit()

# --- Implementation of the Current Plan ---
# I will filter the transactions for 'Rafa_AI' in February 2023 (day_of_year from 32 to 59)
# that meet all the conditions of fee rule ID=17: `card_scheme` must be 'SwiftCharge',
# `is_credit` must be True, and `aci` must be 'A'. Then, for these matching transactions,
# I will calculate the total fee difference using the formula:
# (99 - 60) * SUM(eur_amount) / 10000.

print("--- Calculating fee difference for 'Rafa_AI' in Feb 2023 under specific conditions ---")

# Step 1: Define the conditions for filtering based on the plan.
# February 2023 corresponds to day_of_year 32 to 59 (2023 is not a leap year).
feb_start_day = 32
feb_end_day = 59
merchant_name = 'Rafa_AI'
required_card_scheme = 'SwiftCharge'
required_is_credit = True
required_aci = 'A'
```

```
# Step 2: Filter the payments DataFrame to find transactions matching all conditions.
matching_transactions = payments_df[
    (payments_df['merchant'] == merchant_name) &
    (payments_df['day_of_year'] >= feb_start_day) &
    (payments_df['day_of_year'] <= feb_end_day) &
    (payments_df['card_scheme'] == required_card_scheme) &
    (payments_df['is_credit'] == required_is_credit) &
    (payments_df['aci'] == required_aci)
]

print(f"\n[INFO] Found {len(matching_transactions)} transactions for '{merchant_name}' in February 2023
↪  matching all specified conditions.")

# Step 3: Calculate the total fee difference for these matching transactions.
if not matching_transactions.empty:
    # Sum the 'eur_amount' for all matching transactions.
    total_matching_volume = matching_transactions['eur_amount'].sum()
    print(f"[INFO] Total volume of these transactions: {total_matching_volume:.2f} EUR")

    # Apply the fee difference formula provided in the plan.
    # Formula: (new_rate - old_rate) * SUM(eur_amount) / 10000
    rate_difference = 99 - 60
    total_fee_difference = (rate_difference * total_matching_volume) / 10000

    # --- Final Conclusion ---
    print("\n---------------------CONCLUSION---------------------")
    print(f"The total fee difference for the matching transactions is: {total_fee_difference:.4f} EUR.")

else:
    # --- Final Conclusion ---
    print("\n---------------------CONCLUSION---------------------")
    print("No transactions were found that match all the specified criteria.")
    print("Therefore, the total fee difference is 0.00 EUR.")
```

Listing 34: **Round 3: Implementation.**

```
f'''Insufficient'''
```

Listing 35: **Round 3: Validation.**

```
f'''Step 3 is wrong'''
```

Listing 36: **Round 3: Routing.** Remove Step 3 from the current plan and add Step 3 again.

```
# Step 1
f'''First, I will filter the `payments.csv` data to select only the
↪  transactions for the merchant 'Rafa_AI' that occurred in February
↪  2023. Since 2023 is not a leap year, February corresponds to the
↪  days of the year from 32 to 59.'''

# Step 2
f'''I will calculate the original total fee paid by Rafa_AI in February
↪  2023. This involves loading the `merchant_data.json` and `fees.json`
↪  files, enriching the transaction data with all attributes required
↪  for fee matching (including merchant details, monthly volume,
↪  monthly fraud rate, and intracountry status), then applying the
↪  corresponding fee rule to each transaction and summing the
↪  calculated fees..'''

# Step 3
f'''Calculate the fee delta by subtracting the original `rate` of fee ID
↪  17 from the new `rate` of 99, multiplying the result by the total
↪  volume of the affected transactions (686.48 EUR), and then dividing
↪  by 10000.'''
```

Listing 37: **Round 4: Plan.**

```
import pandas as pd
import json
```

```
# --- 1. Load All Necessary Data ---
# This part is based on the loading logic from the base code.
try:
    payments_df = pd.read_csv('data/payments.csv')
    merchant_data_df = pd.read_json('data/merchant_data.json')
    fees_df = pd.read_json('data/fees.json')
except FileNotFoundError as e:
    print(f"Error loading data: {e}. Please ensure all data files are in the 'data/' directory.")
    exit()

# --- Implementation of the Current Plan ---
# Calculate the fee delta by subtracting the original `rate` of fee ID 17 from the new `rate` of 99,
# multiplying the result by the total volume of the affected transactions (686.48 EUR),
# and then dividing by 10000.

print("--- Calculating fee delta for affected 'Rafa_AI' transactions ---")

# Step 1: Define the constants given in the plan.
new_rate = 99
total_affected_volume = 686.48  # EUR
fee_divisor = 10000

# Step 2: Find the original rate for fee ID 17 from the fees_df DataFrame.
fee_rule_17 = fees_df[fees_df['ID'] == 17]

if fee_rule_17.empty:
    print("\n[ERROR] Fee rule with ID 17 was not found in fees.json.")
    original_rate = None # Set to None to prevent further calculation
else:
    # Extract the 'rate' value from the filtered DataFrame.
    # .iloc[0] selects the first row of the result.
    original_rate = fee_rule_17['rate'].iloc[0]
    print(f"\n[INFO] Original rate for fee ID 17 is: {original_rate}")
    print(f"[INFO] New rate to be applied is: {new_rate}")
    print(f"[INFO] Total volume of affected transactions is: {total_affected_volume:.2f} EUR")

# Step 3: Calculate the fee delta using the specified formula.
if original_rate is not None:
    # Calculate the difference in rates.
    rate_difference = new_rate - original_rate

    # Apply the formula: (rate_difference * total_volume) / 10000
    fee_delta = (rate_difference * total_affected_volume) / fee_divisor

    # --- Final Conclusion ---
    print("\n---------------------CONCLUSION---------------------")
    print(f"The calculated fee delta, by changing the rate from {original_rate} to {new_rate} for the
↪  affected volume, is: {fee_delta:.4f} EUR.")

else:
    # --- Final Conclusion ---
    print("\n---------------------CONCLUSION---------------------")
    print("Could not perform calculation because the original fee rate for ID 17 was not found.")
```

Listing 38: **Round 4: Implementation.**

```
f'''Insufficient'''
```

Listing 39: **Round 4: Validation.**

```
f'''Add Step'''
```

Listing 40: **Round 4: Routing.** Add Step 4 to the current plan.

```
# Step 1
f'''First, I will filter the `payments.csv` data to select only the
↪  transactions for the merchant 'Rafa_AI' that occurred in February
↪  2023. Since 2023 is not a leap year, February corresponds to the
↪  days of the year from 32 to 59.'''

# Step 2
```

```
2700   f'''I will calculate the original total fee paid by Rafa_AI in February
2701   ↪  2023. This involves loading the `merchant_data.json` and `fees.json`
2702   ↪  files, enriching the transaction data with all attributes required
2703   ↪  for fee matching (including merchant details, monthly volume,
2704   ↪  monthly fraud rate, and intracountry status), then applying the
2705   ↪  corresponding fee rule to each transaction and summing the
2706   ↪  calculated fees..'''

2707   # Step 3
2708   f'''Calculate the fee delta by subtracting the original `rate` of fee ID
2709   ↪  17 from the new `rate` of 99, multiplying the result by the total
2710   ↪  volume of the affected transactions (686.48 EUR), and then dividing
2711   ↪  by 10000.'''

2712   # Step 4
2713   f'''Calculate the total original fee for Rafa_AI's transactions in
2714   ↪  February 2023 that are affected by fee ID=17, and then calculate the
2715   ↪  new total fee for the same transactions using the modified rate of
2716   ↪  99. The delta is the difference between these two amounts.'''
```

Listing 41: **Round 5: Plan.**

```python
import pandas as pd
import json

# --- 1. Load All Necessary Data ---
# This part is based on the loading logic from the base code.
try:
    payments_df = pd.read_csv('data/payments.csv')
    merchant_data_df = pd.read_json('data/merchant_data.json')
    fees_df = pd.read_json('data/fees.json')
except FileNotFoundError as e:
    print(f"Error loading data: {e}. Please ensure all data files are in the 'data/' directory.")
    exit()

# --- Implementation of the Current Plan ---
# Calculate the total original fee for Rafa_AI's transactions in February 2023
# that are affected by fee ID=17, and then calculate the new total fee for the same
# transactions using the modified rate of 99. The delta is the difference.

print("--- Identifying transactions for 'Rafa_AI' in February 2023 affected by fee rule ID 17 ---")

# Step 1: Filter payments for 'Rafa_AI' in February 2023.
# February corresponds to days of the year from 32 to 59 (since 2023 is not a leap year).
feb_transactions_df = payments_df[
    (payments_df['merchant'] == 'Rafa_AI') &
    (payments_df['day_of_year'] >= 32) &
    (payments_df['day_of_year'] <= 59)
].copy()

# Step 2: Enrich transaction data with merchant-specific information.
# Get the specific data for 'Rafa_AI' from the merchant data file.
rafa_ai_data = merchant_data_df[merchant_data_df['merchant'] == 'Rafa_AI']

# Merge the February transactions with the merchant's data.
if not rafa_ai_data.empty and not feb_transactions_df.empty:
    # Use a left merge to add merchant details (like mcc, account_type) to each transaction.
    full_transaction_data = feb_transactions_df.merge(rafa_ai_data, on='merchant', how='left')
else:
    print("\n[ERROR] No transactions found for 'Rafa_AI' in February or 'Rafa_AI' data not found.")
    exit()

# Step 3: Retrieve the specific conditions for the fee rule with ID=17.
fee_rule_17 = fees_df[fees_df['ID'] == 17]

if fee_rule_17.empty:
    print("\n[ERROR] Fee rule with ID 17 was not found in fees.json.")
    exit()
else:
    # Use .iloc[0] to get the rule as a Series
    fee_rule_17 = fee_rule_17.iloc[0]

# Step 4: Filter the transactions that match all conditions of the fee rule.
# The manual states that a null or empty list in a rule field means it applies to all values.
mask = pd.Series(True, index=full_transaction_data.index)

# Match card_scheme (string)
if fee_rule_17['card_scheme']:
    mask &= (full_transaction_data['card_scheme'] == fee_rule_17['card_scheme'])

# Match account_type (list)
```

```
if fee_rule_17['account_type']: # An empty list evaluates to False, correctly skipping the filter
    mask &= full_transaction_data['account_type'].isin(fee_rule_17['account_type'])

# Match merchant_category_code (list)
if fee_rule_17['merchant_category_code']:
    mask &= full_transaction_data['merchant_category_code'].isin(fee_rule_17['merchant_category_code'])

# Match aci (list)
if fee_rule_17['aci']:
    mask &= full_transaction_data['aci'].isin(fee_rule_17['aci'])

# Match is_credit (boolean, but can be null)
if fee_rule_17['is_credit'] is not None:
    mask &= (full_transaction_data['is_credit'] == fee_rule_17['is_credit'])

# Apply the combined mask to get the affected transactions
affected_transactions_df = full_transaction_data[mask]

# Step 5: Calculate the original and new fees for the affected transactions.
if affected_transactions_df.empty:
    print("\n[INFO] No transactions for 'Rafa_AI' in February match the criteria for fee rule ID 17.")
    total_original_fee = 0
    total_new_fee = 0
    fee_delta = 0
else:
    total_affected_volume = affected_transactions_df['eur_amount'].sum()
    num_affected_transactions = len(affected_transactions_df)

    print(f"\n[INFO] Found {num_affected_transactions} transactions matching the fee rule.")
    print(f"[INFO] Total volume of these transactions: {total_affected_volume:.2f} EUR")

    # Get fee parameters from the rule
    original_rate = fee_rule_17['rate']
    fixed_amount = fee_rule_17['fixed_amount']
    new_rate = 99
    fee_divisor = 10000

    # Calculate total original fee using the formula: fee = fixed_amount + rate * value / 10000
    # Vectorized calculation for efficiency
    total_fixed_component = fixed_amount * num_affected_transactions
    total_original_rate_component = (original_rate * total_affected_volume) / fee_divisor
    total_original_fee = total_fixed_component + total_original_rate_component

    # Calculate total new fee with the modified rate
    total_new_rate_component = (new_rate * total_affected_volume) / fee_divisor
    total_new_fee = total_fixed_component + total_new_rate_component

    # The delta is the difference between the new and original total fees
    fee_delta = total_new_fee - total_original_fee

# --- Final Conclusion ---
print("\n---------------------CONCLUSION----------------------")
print(f"Total original fee for affected transactions: {total_original_fee:.4f} EUR")
print(f"Total new fee with rate={new_rate}: {total_new_fee:.4f} EUR")
print(f"The calculated fee delta is: {fee_delta:.4f} EUR.")
print("This represents the total increase in fees for Rafa_AI's affected February transactions due to the
↪   rate change.")
```

Listing 42: **Round 5: Implementation.**

```
f'''Sufficient'''
```

Listing 43: **Round 4: Validation.**

```
import pandas as pd
import json

# --- 1. Load All Necessary Data ---
# This part is based on the loading logic from the base code.
payments_df = pd.read_csv('data/payments.csv')
merchant_data_df = pd.read_json('data/merchant_data.json')
fees_df = pd.read_json('data/fees.json')

# --- Implementation of the Calculation ---
# Calculate the total original fee for Rafa_AI's transactions in February 2023
# that are affected by fee ID=17, and then calculate the new total fee for the same
# transactions using the modified rate of 99. The delta is the difference.

# Step 1: Filter payments for 'Rafa_AI' in February 2023.
# February corresponds to days of the year from 32 to 59 (since 2023 is not a leap year).
feb_transactions_df = payments_df[
    (payments_df['merchant'] == 'Rafa_AI') &
    (payments_df['day_of_year'] >= 32) &
    (payments_df['day_of_year'] <= 59)
```

```
].copy()

# Step 2: Enrich transaction data with merchant-specific information.
# Get the specific data for 'Rafa_AI' from the merchant data file.
rafa_ai_data = merchant_data_df[merchant_data_df['merchant'] == 'Rafa_AI']

# Merge the February transactions with the merchant's data.
full_transaction_data = pd.DataFrame()
if not rafa_ai_data.empty and not feb_transactions_df.empty:
    # Use a left merge to add merchant details (like mcc, account_type) to each transaction.
    full_transaction_data = feb_transactions_df.merge(rafa_ai_data, on='merchant', how='left')

# Step 3: Retrieve the specific conditions for the fee rule with ID=17.
fee_delta = 0.0
if not full_transaction_data.empty:
    fee_rule_17 = fees_df[fees_df['ID'] == 17]

    if not fee_rule_17.empty:
        # Use .iloc[0] to get the rule as a Series
        fee_rule_17 = fee_rule_17.iloc[0]

        # Step 4: Filter the transactions that match all conditions of the fee rule.
        # The manual states that a null or empty list in a rule field means it applies to all values.
        mask = pd.Series(True, index=full_transaction_data.index)

        # Match card_scheme (string)
        if fee_rule_17['card_scheme']:
            mask &= (full_transaction_data['card_scheme'] == fee_rule_17['card_scheme'])

        # Match account_type (list)
        if fee_rule_17['account_type']: # An empty list evaluates to False, correctly skipping the filter
            mask &= full_transaction_data['account_type'].isin(fee_rule_17['account_type'])

        # Match merchant_category_code (list)
        if fee_rule_17['merchant_category_code']:
            mask &=
            ↪  full_transaction_data['merchant_category_code'].isin(fee_rule_17['merchant_category_code'])

        # Match aci (list)
        if fee_rule_17['aci']:
            mask &= full_transaction_data['aci'].isin(fee_rule_17['aci'])

        # Match is_credit (boolean, but can be null)
        if fee_rule_17['is_credit'] is not None:
            mask &= (full_transaction_data['is_credit'] == fee_rule_17['is_credit'])

        # Apply the combined mask to get the affected transactions
        affected_transactions_df = full_transaction_data[mask]

        # Step 5: Calculate the original and new fees for the affected transactions.
        if not affected_transactions_df.empty:
            total_affected_volume = affected_transactions_df['eur_amount'].sum()
            num_affected_transactions = len(affected_transactions_df)

            # Get fee parameters from the rule
            original_rate = fee_rule_17['rate']
            fixed_amount = fee_rule_17['fixed_amount']
            new_rate = 99
            fee_divisor = 10000

            # Calculate total original fee using the formula: fee = fixed_amount + rate * value / 10000
            total_fixed_component = fixed_amount * num_affected_transactions
            total_original_rate_component = (original_rate * total_affected_volume) / fee_divisor
            total_original_fee = total_fixed_component + total_original_rate_component

            # Calculate total new fee with the modified rate
            total_new_rate_component = (new_rate * total_affected_volume) / fee_divisor
            total_new_fee = total_fixed_component + total_new_rate_component

            # The delta is the difference between the new and original total fees
            fee_delta = total_new_fee - total_original_fee

# --- Final Output ---
print(f"{fee_delta:.14f}")
```

Listing 44: **Final solution code.**

```
f'''2.67727200000000'''
```

Listing 45: **Generated answer by DS-STAR.** Obtained by executing the above final solution code.

# H PROMPTS

## H.1 ANALYZER AGENT

```
f'''You are an expert data analysist.
Generate a Python code that loads and describes the content of
↪  {filename}.

# Requirement
- The file can both unstructured or structured data.
- If there are too many structured data, print out just few examples.
- Print out essential informations. For example, print out all the
↪   column names.
- The Python code should print out the content of {filename}.
- The code should be a single-file Python program that is self-contained
↪   and can be executed as-is.
- Your response should only contain a single code block.
- Important: You should not include dummy contents since we will debug
↪   if error occurs.
- Do not use try: and except: to prevent error. I will debug it
↪   later.'''
```

Listing 46: Prompt used to generate Python scripts that describe data files.

DS-STAR begins by creating a description of each data file. Specifically, it utilizes an analyzer agent $\mathcal{A}_{\text{analyzer}}$ that takes a file name (*e.g.*, 'payment.csv') as input to generate a Python script $s_{\text{desc}}$, using the above prompt. The script is then executed to summarize the dataset's core information. An example of this process, including the generated script and its output, is provided in Appendix D.

## H.2 PLANNER AGENT

```
f'''You are an expert data analysist.
In order to answer factoid questions based on the given data, you have
↪  to first plan effectively.

# Question
{question}

# Given data: {filenames}
{filenames #1}
{summaries #1}
...
{filenames #N}
{summaries #N}

# Your task
- Suggest your very first step to answer the question above.
- Your first step does not need to be sufficient to answer the question.
- Just propose a very simple initial step, which can act as a good
↪   starting point to answer the question.
- Your response should only contain an initial step.'''
```

Listing 47: Prompt used to generate an initial plan.

DS-STAR begins the solution generation process after generating analytic descriptions of $N$ data files. First, a planner agent $\mathcal{A}_{\text{planner}}$ generates an *initial* high-level executable step $p_0$ using the above prompt. Here, the query $q$ and obtained data descriptions are used as inputs.

```
f'''You are an expert data analysist.
In order to answer factoid questions based on the given data, you have
↪  to first plan effectively.
Your task is to suggest next plan to do to answer the question.

# Question
{question}
```

```
# Given data: {filenames}
{filenames #1}
{summaries #1}
...
{filenames #N}
{summaries #N}

# Current plans
1. {Step 1}
...
k. {Step k}

# Obtained results from the current plans:
{result}

# Your task
- Suggest your next step to answer the question above.
- Your next step does not need to be sufficient to answer the question,
↪  but if it requires only final simple last step you may suggest it.
- Just propose a very simple next step, which can act as a good
↪  intermediate point to answer the question.
- Of course your response can be a plan which could directly answer the
↪  question.
- Your response should only contain an next step without any
↪  explanation.'''
```

Listing 48: Prompt used to generate a plan after the initialization step.

After the initial plan is obtained, DS-STAR's planner agent generates a subsequent step using the above prompt. Unlike the initialization round, planner agent $\mathcal{A}_{\texttt{analyzer}}$ additionally uses the intermediate step-by-step plans, and the obtained results after execution of its implementation.

### H.3 CODER AGENT

```
f'''# Given data: {filenames}
{filenames #1}
{summaries #1}
...
{filenames #N}
{summaries #N}

# Plan
{plan}

# Your task
- Implement the plan with the given data.
- Your response should be a single markdown Python code (wrapped in
↪  ```).
- There should be no additional headings or text in your response.'''
```

Listing 49: Prompt used to generate a implementation of an initial plan in a Python code.

The initial step for the plan is implemented as a code script $s_0$ by a coder agent $\mathcal{A}_{\texttt{coder}}$, using the above prompt. Here, analytic descriptions of data files are additionally used to provide some essential information like column names.

```
f'''You are an expert data analysist.
Your task is to implement the next plan with the given data.

# Given data: {filenames}
{filenames #1}
{summaries #1}
...
{filenames #N}
```

```
{summaries #N}

# Base code
```python
{base_code}
```

# Previous plans
1. {Step 1}
...
k. {Step k}

# Current plan to implement
{Step k+1}

# Your task
- Implement the current plan with the given data.
- The implementation should be done based on the base code.
- The base code is an implementation of the previous plans.
- Your response should be a single markdown Python code (wrapped in
↪   ```).
- There should be no additional headings or text in your response.'''
```

Listing 50: Prompt used to generate a implementation of a plan after the initialization round.

For every round beyond the initial one, DS-STAR's coder agent $\mathcal{A}_{\texttt{coder}}$ converts the current plan step $p_{k+1}$ into a Python script using the above prompt. Unlike the initial round, the agent is provided with the intermediate solution code that implements all preceding step $p_0, \cdots, p_k$. This allows the agent to incrementally build upon the existing code, simplifying the implementation of each subsequent step.

## H.4 VERIFIER AGENT

```
f'''You are an expert data analysist.
Your task is to check whether the current plan and its code
↪   implementation is enough to answer the question.
# Plan
1. {Step 1}
...
k. {Step k}

# Code
```python
{code}
```

# Execution result of code
{result}

# Question
{question}

# Your task
- Verify whether the current plan and its code implementation is enough
↪   to answer the question.
- Your response should be one of 'Yes' or 'No'.
- If it is enough to answer the question, please answer 'Yes'.
- Otherwise, please answer 'No'.'''
```

Listing 51: Prompt used to verify the current plan.

We introduce a verifier agent $\mathcal{A}_{\texttt{verifier}}$. At any given round, this agent evaluates the state of the solution, *i.e.*, whether the plan in sufficient or insufficient to solve the problem. The evaluation is

based on the cumulative plan, the user's query, the solution code, which is an implementation of the cumulative plan, and its execution result, using the above prompt.

## H.5   ROUTER AGENT

```
f'''You are an expert data analysist.
Since current plan is insufficient to answer the question, your task is
↪   to decide how to refine the plan to answer the question.

# Question
{question}

# Given data: {filenames}
{filenames #1}
{summaries #1}
...
{filenames #N}
{summaries #N}

# Current plans
1. {Step 1}
...
k. {Step k}

# Obtained results from the current plans:
{result}

# Your task
- If you think one of the steps of current plans is wrong, answer among
↪   the following options: Step 1, Step 2, ..., Step K.
- If you think we should perform new NEXT step, answer as 'Add Step'.
- Your response should only be Step 1 - Step K or Add Step.'''
```

Listing 52: Prompt used for the router agent which determines how to refine the plan.

If the verifier agent $\mathcal{A}_{\texttt{verifier}}$ determines that the current plan is insufficient to solve the user's query, DS-STAR must decide how to proceed. To this end, DS-STAR employs a router agent $\mathcal{A}_{\texttt{router}}$ which decides whether to append a new step or to correct an existing one, which leverages the above prompt.

## H.6   FINALYZER AGENT

```
f'''You are an expert data analysist.
You will answer factoid question by loading and referencing the
↪   files/documents listed below.
You also have a reference code.
Your task is to make solution code to print out the answer of the
↪   question following the given guideline.

# Given data: {filenames}
{filenames #1}
{summaries #1}
...
{filenames #N}
{summaries #N}

# Reference code
```python{code}
```

# Execution result of reference code
{result}

# Question
{question}

# Guidelines
```

```
{guidelines}

# Your task
- Modify the solution code to print out answer to follow the give
↪  guidelines.
- If the answer can be obtained from the execution result of the
↪  reference code, just generate a Python code that prints out the
↪  desired answer.
- The code should be a single-file Python program that is self-contained
↪  and can be executed as-is.
- Your response should only contain a single code block.
- Do not include dummy contents since we will debug if error occurs.
- Do not use try: and except: to prevent error. I will debug it later.
- All files/documents are in `data/` directory.'''
```

Listing 53: Prompt used to generate a final solution, which outputs the answer with the desired format.

To ensure properly formatted output, DS-STAR employs a finalyzer agent $\mathcal{A}_{\texttt{finalyzer}}$, which takes formatting guidelines, if exist, and generates the final solution code.

## H.7 DEBUGGING AGENT

```
f'''# Error report
{bug}

# Your task
- Remove all unnecessary parts of the above error report.
- We are now running {filename}.py. Do not remove where the error
↪  occurred.'''
```

Listing 54: Prompt used to summarize the traceback of the error.

When debugging, we first summarize the obtained traceback of error using the above prompt, since the traceback may be too lengthy.

```
f'''# Code with an error:
```python
{code}
```

# Error:
{bug}

# Your task
- Please revise the code to fix the error.
- Provide the improved, self-contained Python script again.
- There should be no additional headings or text in your response.
- Do not include dummy contents since we will debug if error occurs.
- All files/documents are in `data/` directory.'''
```

Listing 55: Prompt used for debugging when analyzing data files.

When generating $\{d_i\}_{i=1}^N$ using $s_{\texttt{desc}}$ obtained from $\mathcal{A}_{\texttt{analyzer}}$, our debugging agent iteratively update the script using only the summarized traceback, using the above prompt.

```
f'''# Given data: {filenames}
{filenames #1}
{summaries #1}
...
{filenames #N}
{summaries #N}

# Code with an error:
```python
```

```
{code}
```

# Error:
{bug}

# Your task
- Please revise the code to fix the error.
- Provide the improved, self-contained Python script again.
- Note that you only have {filenames} available.
- There should be no additional headings or text in your response.
- Do not include dummy contents since we will debug if error occurs.
- All files/documents are in `data/` directory.'''
```

Listing 56: Prompt used for debugging when generating a solution code.

Once DS-STAR obtains $\{d_i\}_{i=1}^{N}$, our debugging agent utilizes such information with the above prompt when generating a solution Python script.

## I  LARGE LANGUAGE MODEL USAGE FOR WRITING

In this paper, Gemini was utilized as a general-purpose writing tool. When draft text was provided to Gemini, it improved the writing (*e.g.*, by correcting grammatical errors). The generated text was then manually revised by the authors.

## J  IN-DEPTH ANALYSIS OF THE VERIFIER AGENT

### J.1  SENSITIVITY ANALYSIS VARYING LLM

Table 10: Sensitivity analysis varying the underlying LLM used for the verifier agent.

| Framework | Model for the verifier agent | Easy-level accuracy | Hard-level accuracy |
|---|---|---|---|
| Model-only | - | 66.67 | 12.70 |
| Amity DA Agent | - | 80.56 | 41.01 |
| DS-STAR (Variant) | Llama 3.1 8B | 83.33 | 42.59 |
| DS-STAR | Gemini-2.5-Pro | 87.50 | 45.24 |

Here, we substitute the base LLM of DS-STAR's verifier agent from Gemini-2.5-Pro to Llama 3.1 8B, which is open-sourced and much more lightweight compared to Gemini-2.5-Pro. The rest of the configurations are set as the same as in our current manuscript (*i.e.*, other sub-agents, such as the planner agent, still use Gemini-2.5-Pro as the base LLM).

As shown in Table 10, we found that using a less capable LLM for the verifier agent leads to a marginal performance decrease on the DABStep benchmark. However, it still shows better performance than Model-only (which achieved a hard-level accuracy of 12.70% using Gemini-2.5-Pro) and also outperforms the best baseline (Amity DA Agent). This indicates that small, open-sourced models like Llama 3.1 also possess the capability to effectively verify plan sufficiency. Therefore, DS-STAR's verifier agent is generalizable and robust regarding the LLM choice.

### J.2  SENSITIVITY ANALYSIS VARYING THE SPECIFIC PROMPT TEMPLATE

Table 11: Sensitivity analysis varying the specific prompt template used to instruct the verifier agent.

| Framework | Input for the verifier agent's prompt | Easy-level accuracy | Hard-level accuracy |
|---|---|---|---|
| DS-STAR (Variant) | User's Query, Plan | 83.33 | 43.92 |
| DS-STAR | User's Query, Plan, Code, Execution result of the code | 87.50 | 45.24 |

Here, we investigate the sensitivity of the prompt design for DS-STAR's verifier agent. Specifically, in our original setup (see Listing 51), the verifier agent takes the user's query, the current plan, the current code, and the corresponding execution result as inputs. To analyze prompt sensitivity, we removed the code and its execution result and instead guided the verifier agent to check the sufficiency of the plan compared to the user's query. Here, we used Gemini-2.5-Pro for all sub-agents in this analysis.

As shown in Table 11, we found that additional information such as the code and its execution results, is indeed helpful for the verifier agent, leading to consistently better performance on the DABStep benchmark. However, even without these extra inputs, the verifier agent still works quite well, showing only a small degradation in performance. This result indicates that the design of the verifier agent's prompt template is not sensitive (*i.e.*, does not significantly impact overall performance). Moreover, we would like to emphasize that the prompt is not overfitted to the benchmark we used, since information about the benchmark is not needed for the input, and therefore it is well generalizable to any kinds of data science tasks.

## K   DS-STAR WITH OPEN-SOURCED LLMS

Table 12: DS-STAR with DeepSeek-V3.

| Framework | Model | Easy-level accuracy | Hard-level accuracy |
|---|---|---|---|
| ReAct | DeepSeek-V3 | 66.67 | 5.56 |
| Open Data Scientist | DeepSeek-V3 | **84.72** | 16.40 |
| **DS-STAR** | DeepSeek-V3 | 79.17 | **28.57** |

Here, we conducted an additional experiment: running DS-STAR with DeepSeek-V3 on the DAB-Step benchmark. As shown in Table 12, our framework is also effective with open-source LLMs, demonstrating a significant outperformance on hard-level tasks over DeepSeek-based frameworks on the validated leaderboard of the DABStep benchmark. This result indicates that our framework is generalizable with respect to the choice of LLM.

