# OpenReview forum: "DS-STAR: Data Science Agent via Iterative Planning and Verification"
_ICLR.cc/2026/Conference — Submitted to ICLR 2026_

### Official Review · Reviewer_Rp7p · 2025-10-19

**Soundness:** 3
**Presentation:** 4
**Contribution:** 1
**Rating:** 2
**Confidence:** 4

**Summary:**

This paper aims to develop data science agents with LLMs. This paper proposes DS-STAR, consisting of (1) data analysis agent to provide description of the given data; (2) a planning-verification-refinement-execution pipeline to solve the problem; (3) debugging agent to debug code; (4) retriever to retrieve relevant data files. This paper conducts extensive experiments on several data science benchmarks: DABStep, KarmaBench, and DA-Code.

**Strengths:**

- The writing is of good-quality and easy to follow.
- The investigated research problem – data science agents – are of significance.
- Extensive ablation studies are provided to verify the effectiveness of the proposed agent framework.

**Weaknesses:**

- Although the research problem is interesting, the technical solution turns out to be plain and naïve. The proposed DA-STAR is lack of technical depths. All the components are well-known recipes for LLM agent community. There are a large number of existing papers that already discuss the planning-execution pipeline. Also, the proposed debugging agent and retriever are also widely investigated for coding agents. As such, I believe the novelty of this paper is quite limited, which is clearly under the bar of top-tier conferences like ICLR.

- The performance improvement is marginal as shonw in Table 2 and Table 3. DS-STAR fails to consistently outperform all the baselines in all the settings, which further decrease the contribution of this paper.

- No empirical results on DA-STAR with open-sourced LLMs. The reliance on commercial LLMs further decrease the contribution of this paper.

**Questions:**

I think this paper does not make substantial contribution to the community. I believe the rebuttal phase cannot resolve the fundamental limitation of this paper.

---

> ### Author Response · Authors · 2025-11-22
>
> Dear Reviewer Rp7p,
>
> We deeply appreciate your invaluable time and efforts in reviewing our manuscript. We respond to each of your comments one-by-one in what follows. The revised manuscript has been marked in blue.
>
> ---
> **[W1] Lack of technical depth.**
>
> First, thanks for acknowledging that the research topic is interesting. We would also like to emphasize that the problem itself represents a significant contribution, since handling heterogeneous and unstructured data formats (not just CSV) is an important real-world challenge that remains a relatively underexplored area.
>
> On the other hand, we respectfully disagree that our DS-STAR lacks technical depth. First of all, as you mentioned, our data file analysis module is a key novelty not found in previous works. Specifically, DS-STAR generates code to analyze and summarize diverse data formats before planning. This step is critical for handling real-world data science tasks and, as our ablation study (see Table 4 in our manuscript) shows, removing it causes hard-level accuracy to degrade from 45.24% to 26.98%.
>
> Moreover, we also argue that our iterative plan generation and verification loop is not a plain and naive pipeline. Specifically, our verifier agent is a novel component that explicitly judges plan sufficiency for open-ended tasks, rather than just code execution success. To further alleviate your concern, we conducted an additional experiment to validate the effectiveness of our verifier agent. Specifically, we compare our approach against a baseline where we generate a full plan at once and use the success of the code execution as the sole plan verifier module, rather than using the step-by-step planning guided by our verifier agent. As shown in the table below, step-by-step planning by checking the sufficiency using the LLM as a judge is the key component enabling the superior performance of our DS-STAR, improving hard-level accuracy from 34.66% to 45.24%. This result clearly indicates that our verifier agent is a novel and effective component of our framework.
>
> Furthermore, our router agent introduces a non-trivial decision-making step, enabling the agent to backtrack and fix erroneous steps rather than merely accumulating flawed ones. This component enables dynamic planning rather than monotonic planning. It also improves performance, by increasing the hard-level accuracy of the DABStep benchmark from 39.95% to 45.24%, as shown in Table 4 of our main manuscript.
>
> Finally, we would like to note that while each component can be seen as a common standard, orchestrating them effectively to multi-agent systems for specific tasks is often non-trivial and is frequently recognized as a significant contribution. For example, SE-Agent [1] also leverages mechanisms like reflection, verification, and refinement as core modules, and it orchestrates them to perform well on the SWE-bench. Additionally, CGI [2] adopts an actor-critique framework, which is similar to a planner-verifier framework of ours, for an effective multi-agent system. We would like to emphasize that the contribution of both of these works lies highly in the effective orchestration of standard techniques, recognized as acceptances by the most recent NeurIPS 2025. We believe that our DS-STAR’s contribution and novelty similarly reside in the effective orchestration of its sub-agents, which we believe as a significant contribution to the ICLR community.
>
> \begin{array}{lccc}
> \hline
> \text{Framework} & \text{Verifier agent} & \text{Easy-level accuracy} & \text{Hard-level accuracy}\newline
> \hline
> \textbf{DS-STAR (Variant)} & \text{X} & 83.33 & 34.66\newline
> \textbf{DS-STAR} & \textbf{O} & \textbf{87.50} & \textbf{45.24}\newline
> \hline
> \end{array}
>
> [1] Lin et al., SE-Agent: Self-Evolution Trajectory Optimization in Multi-Step Reasoning with LLM-Based Agents, NeurIPS 2025.
>
> [2] Yang et al., The Lighthouse of Language: Enhancing LLM Agents via Critique-Guided Improvement, NeurIPS 2025.

---

> ### Author Response · Authors · 2025-11-22
>
> **[W2] The performance improvement is marginal as shown in Table 2 and Table 3.**
>
> Thanks for the careful feedback. However, we still believe that our experimental results demonstrate a significant contribution. Specifically, our main claim is that DS-STAR is most effective at handling complex, hard tasks involving heterogeneous data, where current SOTA agents struggle (see Figure 1 of the manuscript).
>
> Specifically, DS-STAR outperforms the SOTA DA-Agent in 4 of the 6 domains in KramaBench (see Table 2) and, most importantly, achieves the highest total score, showing a strong 4.9% absolute improvement.
>
> Moreover, we acknowledge that the 1.5% absolute gain on the total score in DA-Code benchmark (see Table 3) can be seen as marginal. However, DS-STAR’s superiority is particularly highlighted on more complex problems, as demonstrated in the hard difficulty tasks. Here, DS-STAR achieves 37.1%, whereas the next-best DA-Agent achieves 32.0%, showing 5.1% absolute improvement. Moreover, DA-Agent might have been overfit to the DA-Code benchmark since it was proposed by the benchmark paper, as demonstrated by its relatively low performance on other benchmarks (see Table 1 and 2 in our main manuscript).
>
> ---
> **[W3] No empirical results on DS-STAR with open-sourced LLMs.**
>
> Thanks for your valuable feedback. To alleviate your concern, we conducted an additional experiment: running DS-STAR with DeepSeek-V3 on the DABStep benchmark. As shown in the table below, our framework is also effective with open-source LLMs, demonstrating a significant outperformance on hard-level tasks over DeepSeek-based frameworks on the validated leaderboard of the DABStep benchmark. This result indicates that our framework is generalizable with respect to the choice of LLM. We included these results in Appendix K of our revised manuscript.
>
> \begin{array}{lccc}
> \hline
> \text{Framework} & \text{Model} & \text{Easy-level accuracy} & \text{Hard-level accuracy}\newline
> \hline
> \text{ReAct} & \text{DeepSeek-V3} & 66.67 & 5.56\newline
> \text{Open Data Scientist} & \text{DeepSeek-V3} & \textbf{84.72} & 16.40\newline
> \textbf{DS-STAR (Ours)} & \text{DeepSeek-V3} & 79.17 & \textbf{28.57}\newline
> \hline
> \end{array}

---

> > ### Comment · Reviewer_Rp7p · 2025-11-26
> >
> > Thanks for the response.
> > - Regarding technical depth, I choose to keep my opinion, which is also mentioned by Reviewer qKAu.
> >
> > - Regarding the marginal performance, the newly added results with DeepSeek-V3 further verify this. According to the provided table, DS-STAR performs worse than Open Data Scientist in easy-level tasks, while achieving better performance in hard-level tasks.
> >
> > Thus, I would keep my score unchanged.

---

### Official Review · Reviewer_eYcv · 2025-10-25

**Soundness:** 3
**Presentation:** 3
**Contribution:** 2
**Rating:** 6
**Confidence:** 3

**Summary:**

We introduce a data science agent (called DS-STAR), which aims to tackle complex data science tasks automatically. DS-STAR makes three key contributions: exploring and extracting context from diverse data formats, evaluating the sufficiency of the analysis plans, and iteratively refining plans. Experiments on three challenging benchmarks show the effectiveness of DS-STAR.

**Strengths:**

1. Developing LLM-driven agents for automating end-to-end data science pipelines is interesting and can augment human analysts.

2. The proposed DS-STAR is reasonable and can be applied to various data science tasks.

3. DS-STAR achieves substantially better performance over several representative baselines across all three challenging benchmarks.

**Weaknesses:**

1. The reported experimental results are significantly better than the baseline methods. However, after looking at the results on the DABStep leaderboard, AgenticData achieved much higher accuracy on the easy level than DS-STAR. In addition, CambioML DS Agent achieved significantly higher accuracy at both the easy and hard levels than DS-STAR. It would be better to provide some in-depth explanation and analysis.

2. The DS-STAR framework relies on multiple LLM API calls throughout its pipeline. A comparative analysis of its computational cost and latency against the baseline methods should be included to provide a complete assessment of its efficiency.

**Questions:**

A comparative analysis of its computational cost and latency against the baseline methods should be included to provide a complete assessment of its efficiency.

---

> ### Author Response · Authors · 2025-11-22
>
> Dear Reviewer eYcv,
>
> We deeply appreciate your invaluable time and efforts in reviewing our manuscript. We respond to each of your comments one-by-one in what follows. The revised manuscript has been marked in blue.
>
> ---
> **[W1] It would be better to provide some in-depth explanation and analysis compared to AgenticData and CambioML DS Agent.**
>
> First of all, we would like to clarify that at the time of submission, CambioML DS Agent and AgenticData were not included in the validated leaderboard (the date of submission reflects the time the results were submitted to the unvalidated leaderboard, which takes additional time to be validated by the authors of the DABStep benchmark; validated in October as far as we know) of the DABStep benchmark and thus were not direct competitors to our work. However, as per your suggestion, we will include an additional in-depth analysis in the related work section of the final manuscript.
>
> ---
> **[W2, Q1] A comparative analysis of its computational cost and latency.**
>
> We would like to first clarify that we have already quantitatively analyzed the LLM usage cost in the Appendix C of the manuscript (also see the below table). Specifically, we evaluated DS-STAR on 10 development tasks (3 easy / 7 hard) from the DABStep benchmark, and reported the average number of LLM calls, input tokens, and output tokens in Table 6 of the current manuscript.
>
> Specifically, we acknowledge that DS-STAR incurs higher cost due to its increased token usage, which is primarily due to the utilization of comprehensive analytic descriptions of each data file. Nevertheless, this translates to a modest practical cost; using Gemini-2.5-Pro (priced at $1.25/$10.00 per 1M input/output tokens), the expense for DS-STAR averages only $0.23 per data science task. Therefore, we argue this higher cost is a justifiable trade-off, as DS-STAR consistently and significantly achieves better results than the baselines (especially compared with single-shot baselines).
>
> To further alleviate your concern, we additionally evaluated the average wall-clock time on 10 development tasks from the DABstep benchmark. As shown in the table below, we acknowledge that our DS-STAR takes modestly longer than ReAct or DA-Agent. However, we would like to emphasize that our framework significantly outperforms both, and more importantly, hard-level tasks often require more than three minutes to solve even by human-experts, indicating that DS-STAR’s latency is reasonable.
>
> \begin{array}{lcccccc}
> \hline
> \text{Method}&\text{Performance on DABStep (Easy / Hard)} & \text{\\# LLM calls} & \text{\\# Input tokens} & \text{\\# Output tokens} & \text{Cost (\\$)} & \text{Time (sec)}\newline
> \hline
> \text{ReAct}&\text{69.44 / 10.05} & 7.1 & 44691 & 2928 & 0.09 & 72.58\newline
> \text{DA-Agent}&\text{68.06 / 22.49} & 8.8 & 39999 & 4123 & 0.09 & 66.39\newline
> \textbf{DS-STAR (Ours)} & \textbf{87.50 / 45.24} & 12.7 & 154669 & 3373 & 0.23 & 175.76\newline
> \hline
> \end{array}

---

### Official Review · Reviewer_qKAu · 2025-10-25

**Soundness:** 2
**Presentation:** 3
**Contribution:** 2
**Rating:** 4
**Confidence:** 3

**Summary:**

This paper proposes DS-STAR, a new agent framework designed to address data science tasks, with a particular focus on handling heterogeneous data formats (e.g., CSV, JSON, unstructured text) that current methods often struggle with. The DS-STAR agent is structured as a workflow composed of three main modules: (1) a data file analysis module to explore and extract context from diverse data files, (2) a verification module (using an LLM as a judge) to evaluate the sufficiency of the analysis plan, and (3) an iterative planning and refinement module that sequentially builds and optimizes the solution. The authors evaluate their method on three data science benchmarks (DABStep, KramaBench, DA-Code), demonstrating improved performance over existing baselines.

**Strengths:**

1. The paper addresses a practical problem. Real-world data science tasks frequently involve a variety of heterogeneous data sources.
2. The experimental evaluation is relatively thorough. The authors have validated their approach on three benchmarks and included ablation studies to demonstrate the contribution of the different components.

**Weaknesses:**

1. The primary weakness of this paper is its limited novelty. The proposed method essentially defines a specific workflow for a data science agent, but the core components are very similar to those used in many existing agent frameworks. Beyond the data file analysis module, the other components—such as using an LLM for verification or employing an iterative planning/refinement loop—are common, almost standard, techniques in the current agent literature.
2. The experimental evaluation, while comprehensive in accuracy, appears to be incomplete in a critical practical dimension: efficiency. The paper does not present an analysis of the computational time or monetary cost (e.g., API calls) of the proposed method compared to the baselines. Given the multi-module, iterative nature of DS-STAR, it is highly likely that it incurs significantly higher latency and cost.

**Questions:**

N/A

---

> ### Author Response · Authors · 2025-11-22
>
> Dear Reviewer qKAu,
>
> We deeply appreciate your invaluable time and efforts in reviewing our manuscript. We respond to each of your comments one-by-one in what follows. The revised manuscript has been marked in blue.
>
> ---
> **[W1] Limited novelty.**
>
> We would like to first emphasize that the problem itself represents a significant contribution, since handling heterogeneous and unstructured data formats (not just CSV) is an important real-world challenge in data science domains that remains a relatively underexplored area.
>
> On the other hand, we respectfully disagree that our DS-STAR lacks technical depth. First of all, as you mentioned, our data file analysis module is a key novelty not found in previous works. Specifically, DS-STAR generates code to analyze and summarize diverse data formats before planning. This step is critical for handling real-world data science tasks and, as our ablation study (see Table 4 in our manuscript) shows, removing it causes hard-level accuracy to degrade from 45.24% to 26.98%.
>
> Moreover, we also argue that our iterative plan generation and verification loop is not a plain and naive pipeline. Specifically, our verifier agent is a novel component that explicitly judges plan sufficiency for open-ended tasks, rather than just code execution success. To further alleviate your concern, we conducted an additional experiment to validate the effectiveness of our verifier agent. Specifically, we compare our approach against a baseline where we generate a full plan at once and use the success of the code execution as the sole plan verifier module, rather than using the step-by-step planning guided by our verifier agent. As shown in the table below, step-by-step planning by checking the sufficiency using the LLM as a judge is the key component enabling the superior performance of our DS-STAR, improving hard-level accuracy from 34.66% to 45.24%. This result clearly indicates that our verifier agent is a novel and effective component of our framework.
>
> Furthermore, our router agent introduces a non-trivial decision-making step, enabling the agent to backtrack and fix erroneous steps rather than merely accumulating flawed ones. This component enables dynamic planning rather than monotonic planning. It also improves performance, by increasing the hard-level accuracy of the DABStep benchmark from 39.95% to 45.24%, as shown in Table 4 of our main manuscript.
>
> Finally, we would like to note that while each component can be seen as a common standard, orchestrating them effectively to multi-agent systems for specific tasks is often non-trivial and is frequently recognized as a significant contribution. For example, SE-Agent [1] also leverages mechanisms like reflection, verification, and refinement as core modules, and it orchestrates them to perform well on the SWE-bench. Additionally, CGI [2] adopts an actor-critique framework, which is similar to a planner-verifier framework of ours, for an effective multi-agent system. We would like to emphasize that the contribution of both of these works lies highly in the effective orchestration of standard techniques, recognized as acceptances by the most recent NeurIPS 2025. We believe that our DS-STAR’s contribution and novelty similarly reside in the effective orchestration of its sub-agents, which we believe as a significant contribution to the ICLR community.
>
> \begin{array}{lccc}
> \hline
> \text{Framework} & \text{Verifier agent} & \text{Easy-level accuracy} & \text{Hard-level accuracy}\newline
> \hline
> \textbf{DS-STAR (Variant)} & \text{X} & 83.33 & 34.66\newline
> \textbf{DS-STAR} & \textbf{O} & \textbf{87.50} & \textbf{45.24}\newline
> \hline
> \end{array}
>
> [1] Lin et al., SE-Agent: Self-Evolution Trajectory Optimization in Multi-Step Reasoning with LLM-Based Agents, NeurIPS 2025.
>
> [2] Yang et al., The Lighthouse of Language: Enhancing LLM Agents via Critique-Guided Improvement, NeurIPS 2025.

---

> ### Author Response · Authors · 2025-11-22
>
> **[W2] The paper does not present an analysis of the computational time or monetary cost.**
>
> We would like to first clarify that we have already quantitatively analyzed the LLM usage cost in the Appendix C of the manuscript (also see the below table). Specifically, we evaluated DS-STAR on 10 development tasks (3 easy / 7 hard) from the DABStep benchmark, and reported the average number of LLM calls, input tokens, and output tokens in Table 6 of the current manuscript.
>
> Specifically, we acknowledge that DS-STAR incurs higher cost due to its increased token usage, which is primarily due to the utilization of comprehensive analytic descriptions of each data file. Nevertheless, this translates to a modest practical cost; using Gemini-2.5-Pro (priced at $1.25/$10.00 per 1M input/output tokens), the expense for DS-STAR averages only $0.23 per data science task. Therefore, we argue this higher cost is a justifiable trade-off, as DS-STAR consistently and significantly achieves better results than the baselines (especially compared with single-shot baselines).
>
> To further alleviate your concern, we additionally evaluated the average wall-clock time on 10 development tasks from the DABstep benchmark. As shown in the table below, we acknowledge that our DS-STAR takes modestly longer than ReAct or DA-Agent. However, we would like to emphasize that our framework significantly outperforms both, and more importantly, hard-level tasks often require more than three minutes to solve even by human-experts, indicating that DS-STAR’s latency is reasonable.
>
> \begin{array}{lcccccc}
> \hline
> \text{Method}&\text{Performance on DABStep (Easy / Hard)} & \text{\\# LLM calls} & \text{\\# Input tokens} & \text{\\# Output tokens} & \text{Cost (\\$)} & \text{Time (sec)}\newline
> \hline
> \text{ReAct}&\text{69.44 / 10.05} & 7.1 & 44691 & 2928 & 0.09 & 72.58\newline
> \text{DA-Agent}&\text{68.06 / 22.49} & 8.8 & 39999 & 4123 & 0.09 & 66.39\newline
> \textbf{DS-STAR (Ours)} & \textbf{87.50 / 45.24} & 12.7 & 154669 & 3373 & 0.23 & 175.76\newline
> \hline
> \end{array}

---

### Official Review · Reviewer_J4Fr · 2025-11-01

**Soundness:** 2
**Presentation:** 2
**Contribution:** 2
**Rating:** 4
**Confidence:** 3

**Summary:**

This paper introduces DS-STAR, a novel data science agent framework designed to automate complex data science tasks through a robust process of iterative planning and verification. The framework is designed to overcome limitations of existing LLM agents, namely their struggle with heterogeneous data formats and the tendency to generate sub-optimal analysis plans due to the difficulty of verifying plan sufficiency in open-ended tasks.

**Strengths:**

- Robust Handling of Heterogeneous Data: The proposed Data File Analysis Module is crucial for real-world applicability. By automatically exploring and extracting structural and content summaries from various heterogeneous formats (CSV, JSON, unstructured text, etc.), DS-STAR significantly broadens the scope of data science agents beyond the structured data limitation of previous methods.


- Effective Iterative Planning Strategy: The use of sequential planning and iterative refinement, starting with a simple executable step and incrementally building the solution based on verification feedback, effectively mimics the interactive analysis process of a domain expert working in a Jupyter Notebook. This approach is highly beneficial for tackling complex, multi-step analysis tasks involving dependent results.

- Superior Performance on Hard Tasks: DS-STAR achieves SOTA performance across multiple challenging data science benchmarks. Its significant outperformance on "Hard Tasks" that require processing multiple, heterogeneous data files strongly validates the efficacy of its key design components, especially the data analysis and verification modules.

**Weaknesses:**

1. Reliability and Bias of the Verification Module: The performance of DS-STAR is highly dependent on the quality of the LLM Judge's judgment of "plan sufficiency." The paper lacks an in-depth analysis of the Judge's reliability, generalizability, and potential inherent biases across different task difficulties or underlying LLM architectures. Errors in the Judge's assessment could lead the agent down flawed iterative paths or cause it to prematurely accept sub-optimal solutions.

2. Efficiency and Cost Concerns: The iterative planning and multi-step verification process (Planner $\rightarrow$ Coder $\rightarrow$ Judge $\rightarrow$ Refinement) is inherently a multi-turn loop. Compared to single-shot planning or baselines that rely only on code execution success, solving a single task likely requires multiple LLM API calls. The paper must quantify the average inference time and API call count for DS-STAR and discuss the trade-offs in latency and computational cost.

3. Boundary Limitations of Data Summarization: While the analysis module is designed to handle heterogeneous data, the ability of a Python script and LLM to generate an accurate and complete text summary for highly complex, large-scale, or deeply nested unstructured data remains questionable. If the generated summary is incomplete or flawed, subsequent planning will be fundamentally biased from the outset.

**Questions:**

Sensitivity Analysis of the LLM Judge: The quality of the LLM Judge determines the framework's performance ceiling. Please conduct a sensitivity analysis on the Judge agent, including:
       1. varying the underlying LLM used for the Judge (e.g., substituting GPT-4 with a powerful open-source model);
       2. varying the specific prompt template used to instruct the Judge. This analysis is critical for assessing the framework's robustness and practical deployability.

---

> ### Author Response · Authors · 2025-11-22
>
> Dear Reviewer J4Fr,
>
> We deeply appreciate your invaluable time and efforts in reviewing our manuscript. We respond to each of your comments one-by-one in what follows. The revised manuscript has been marked in blue.
>
> ---
> **[W1, Q1] In-depth analysis of the LLM Judge’s judgment of “plan sufficiency”: Sensitivity analysis varying the underlying LLM used for the judge.**
>
> Thanks for your constructive comments. Following your suggestion, we substituted the base LLM of DS-STAR’s verifier agent from Gemini-2.5-Pro to Llama 3.1 8B, which is open-sourced and much more lightweight compared to Gemini-2.5-Pro. The rest of the configurations are set as the same as in our current manuscript (i.e., other sub-agents, such as the planner agent, still use Gemini-2.5-Pro as the base LLM).
>
> As shown in the table below, we found that using a less capable LLM for the verifier agent leads to a marginal performance decrease on the DABStep benchmark. However, it still shows better performance than Model-only (which achieved a hard-level accuracy of 12.70% using Gemini-2.5-Pro) and also outperforms the best baseline (Amity DA Agent). This indicates that small, open-sourced models like Llama 3.1 also possess the capability to effectively verify plan sufficiency. Therefore, DS-STAR’s verifier agent is generalizable and robust regarding the LLM choice. We included these results in Appendix J.1 of our revised manuscript.
>
> \begin{array}{lccc}
> \hline
> \text{Framework}&\text{Model for the verifier agent}&\text{Easy-level accuracy} & \text{Hard-level accuracy}\newline
> \hline
> \text{Model-only}&\text{-}&66.67&12.70\newline
> \text{Amity DA Agent}&\text{-}&80.56&41.01\newline
> \textbf{DS-STAR (Variant)} & \text{Llama 3.1 8B} & 83.33 & 42.59\newline
> \textbf{DS-STAR} & \text{Gemini-2.5-Pro} & 87.50 & 45.24\newline
> \hline
> \end{array}
>
> ---
> **[W1, Q2] In-depth analysis of the LLM Judge’s judgment of “plan sufficiency”: Sensitivity analysis varying the specific prompt template used to instruct the judge.**
>
> Thank you for your constructive comments. Following your suggestion, we investigated the sensitivity of the prompt design for DS-STAR’s verifier agent. Specifically, in our original setup (see Listing 51 in our revised manuscript), the verifier agent takes the user’s query, the current plan, the current code, and the corresponding execution result as inputs. To analyze prompt sensitivity, as shown in the below prompt, we removed the code and its execution result and instead guided the verifier agent to check the sufficiency of the plan compared to the user’s query. Here, we used Gemini-2.5-Pro for all sub-agents in this analysis.
>
> As shown in the table below, we found that additional information such as the code and its execution results, is indeed helpful for the verifier agent, leading to consistently better performance on the DABStep benchmark. However, even without these extra inputs, the verifier agent still works quite well, showing only a small degradation in performance. This result indicates that the design of the verifier agent’s prompt template is not sensitive (i.e., does not significantly impact overall performance). Moreover, we would like to emphasize that the prompt is not overfitted to the benchmark we used, since information about the benchmark is not needed for the input, and therefore it is well generalizable to any kinds of data science tasks (see below verifier prompt). We included these results in Appendix J.2 of our revised manuscript.
>
> \begin{array}{lccc}
> \hline
> \text{Framework}&\text{Input for the verifier agent’s prompt}&\text{Easy-level accuracy} & \text{Hard-level accuracy}\newline
> \hline
> \textbf{DS-STAR (Variant)} & \text{User’s Query, Plan} & 83.33 & 43.92\newline
> \textbf{DS-STAR} & \text{User’s Query, Plan, Code, Execution result of the code} & 87.50 & 45.24\newline
> \hline
> \end{array}
>
> **Modified prompt for the verifier agent.**
> ```
> You are an expert data analysist.
> Your task is to check whether the current plan is enough to answer the question.
>
> # Plan
> 1. {Step 1}
> ...
> k. {Step k}
>
> # Question
> {question}
>
> # Your task
> - Verify whether the current plan is enough to answer the question.
> - Your response should be one of 'Yes' or 'No'.
> - If it is enough to answer the question, please answer 'Yes'.
> - Otherwise, please answer 'No'.
> ```

---

> ### Author Response · Authors · 2025-11-22
>
> **[W2] Efficiency and cost concerns. The paper must quantify the average inference time and API call count for DS-STAR and discuss the trade-offs in latency and computational cost.**
>
> We would like to first clarify that we have already quantitatively analyzed the LLM usage cost in the Appendix C of the manuscript (also see the below table). Specifically, we evaluated DS-STAR on 10 development tasks (3 easy / 7 hard) from the DABStep benchmark, and reported the average number of LLM calls, input tokens, and output tokens in Table 6 of the current manuscript.
>
> Specifically, we acknowledge that DS-STAR incurs higher cost due to its increased token usage, which is primarily due to the utilization of comprehensive analytic descriptions of each data file. Nevertheless, this translates to a modest practical cost; using Gemini-2.5-Pro (priced at $1.25/$10.00 per 1M input/output tokens), the expense for DS-STAR averages only $0.23 per data science task. Therefore, we argue this higher cost is a justifiable trade-off, as DS-STAR consistently and significantly achieves better results than the baselines (especially compared with single-shot baselines).
>
> To further alleviate your concern, we additionally evaluated the average wall-clock time on 10 development tasks from the DABstep benchmark. As shown in the table below, we acknowledge that our DS-STAR takes modestly longer than ReAct or DA-Agent. However, we would like to emphasize that our framework significantly outperforms both, and more importantly, hard-level tasks often require more than three minutes to solve even by human-experts, indicating that DS-STAR’s latency is reasonable.
>
> \begin{array}{lcccccc}
> \hline
> \text{Method}&\text{Performance on DABStep (Easy / Hard)} & \text{\\# LLM calls} & \text{\\# Input tokens} & \text{\\# Output tokens} & \text{Cost (\\$)} & \text{Time (sec)}\newline
> \hline
> \text{ReAct}&\text{69.44 / 10.05} & 7.1 & 44691 & 2928 & 0.09 & 72.58\newline
> \text{DA-Agent}&\text{68.06 / 22.49} & 8.8 & 39999 & 4123 & 0.09 & 66.39\newline
> \textbf{DS-STAR (Ours)} & \textbf{87.50 / 45.24} & 12.7 & 154669 & 3373 & 0.23 & 175.76\newline
> \hline
> \end{array}
>
> ---
> **[W3] Boundary limitations of data summarization. The ability of a Python script and LLM to generate an accurate and complete text summary for highly complex, large-scale or deeply nested unstructured data remains questionable. If the generated summary is incomplete or flawed, subsequent planning will be fundamentally biased.**
>
> Thank you for raising a good point. Indeed, as shown in Table 4 of our main manuscript, the data file summaries are essential for accurate data science tasks. We acknowledge that an incomplete or flawed summary can fundamentally bias subsequent planning and lead to low performance, which is a potential limitation of our current framework. On the other hand, one way to alleviate such issues is to iteratively refine the summarization by providing feedback, obtained from planner agent or code execution, to our analyzer agent. We leave this as the future extension of DS-STAR.
>
> However, we would like to emphasize that the current analyzer agent is sufficiently robust to summarize even highly complex data. For example, our analyzer agent successfully summarizes the file 1-s2.0-S0092867420301070-mmc2.xlsx from the biomedical domain of KramaBench. This file is particularly complex as it contains three separate sheets within one Excel file. As demonstrated in the summarization result shown in Listing 5 of Appendix E of our revised manuscript, our agent successfully reads and analyzes each sheet one-by-one, which qualitatively shows its robustness and capability in handling multi-sheet, complex data structures.
>
> Moreover, to further alleviate your concern, we created a complex Excel file where three tables of different shapes are placed in random positions in one sheet. This data specifically requires DS-STAR to exactly identify the bounding boxes of each table. As demonstrated in Listing 6 of Appendix E of our revised manuscript, DS-STAR’s analyzer agent successfully handles such complex data layouts, which clearly shows its robustness.

---

> > ### Comment · Reviewer_J4Fr · 2025-11-27
> >
> > Thank you for the authors' response, which has clarified some of my questions.
> > DS-STAR is a multi-agent framework, but it is primarily prompt-driven and leans more toward engineering rather than addressing core research problems. Additionally, DS-STAR currently lacks substantial algorithmic or theoretical contributions.
> > So I will keep my score.

---

### Meta-Review · Area_Chair_9atN · 2025-12-20

**Summary:**

The primary concern of reviewers appears to be their perception that this work has limited algorithmic and theoretical contribution.
Reviewers wanted more concrete runtime and API cost estimates.
There is also concern using the LLM as a judge framework.

**Reviewer Concerns:**

The authors point to token and call statistics and estimated costs per task which seems to address the reviewer desires to have efficiency and costs quantified.  Alternative LLMs were swapped to examine the LLM as a judge sensitivity.

**Reviewer Scores:**

Reviewers seems generally unswayed by the rebuttal with multiple reviewers explicitly stating that their scores remain unchanged.
While the AC actually does think this paper is quite interesting with these reviews and the reviewers holding strong to their assessments it is just not possible to justify an acceptance recommendation. The AC recommends rejection.

---

### Decision · Program_Chairs · 2026-01-26

Reject